# NeuralOS: Towards Simulating Operating Systems via Neural Generative Models

**Luke Rivard**[1]    **Sun Sun**[2]    **Hongyu Guo**[2]    **Wenhu Chen**[1]    **Yuntian Deng**[1]

[1]University of Waterloo    [2]National Research Council Canada

{jlrivard, wenhu.chen, yuntian}@uwaterloo.ca
{sun.sun, hongyu.guo}@nrc-cnrc.gc.ca

## Abstract

We introduce NeuralOS, a neural framework that simulates graphical user interfaces (GUIs) of operating systems by directly predicting screen frames in response to user inputs such as mouse movements, clicks, and keyboard events. NeuralOS combines a recurrent neural network (RNN), which tracks computer state, with a diffusion-based neural renderer that generates screen images. The model is trained on a dataset of Ubuntu XFCE recordings, which include both randomly generated interactions and realistic interactions produced by AI agents. Experiments show that NeuralOS successfully renders realistic GUI sequences, accurately captures mouse interactions, and reliably predicts state transitions like application launches. Beyond reproducing existing systems, NeuralOS shows that synthesized training data can teach the model to simulate applications that were never installed, as illustrated by a Doom application, and suggests a path toward learning user interfaces purely from synthetic demonstrations.

## 1 Introduction

> *"Chatting" with LLM feels like using an 80s computer terminal. The GUI hasn't been invented yet, but some properties of it can start to be predicted.*
>
> — **Andrej Karpathy**

Recent breakthroughs in generative models have transformed human-computer interaction, making it increasingly adaptive, personalized, and intuitive. Historically, computing interfaces were rigid and predefined, such as command-line terminals and static graphical menus (Engelbart, 1968). The emergence of large language models (LLMs) and multimodal AI systems expanded this paradigm by enabling interactions through natural language (Radford et al., 2019; Brown et al., 2020), images (Ho et al., 2020; Lipman et al., 2022; Radford et al., 2021; Song et al., 2020b), and videos (Ho et al., 2022b; Singer et al., 2022b; OpenAI, 2024). Recently, generative models have even begun simulating dynamic visual environments (Ha & Schmidhuber, 2018a; He et al., 2025a), notably interactive video games (Feng et al., 2024; Oh et al., 2015; Valevski et al., 2024). These advancements suggest a future where computing interfaces could become fully generative, dynamically adapting in real-time based on user inputs, contexts, and intentions (Deka et al., 2017).

In this work, we study whether an operating system-like interface can be modeled using a neural generative model, rather than a manually programmed system. In doing so, we investigate the modeling and optimization challenges, such as precise cursor control and state tracking. Such a neural interface provides a basis for human-computer interaction research on user-controllable UI generation (e.g., prompt-to-UI personalization (Tolomei et al., 2023), and provides a safe environment in which computer-use agents can be trained and evaluated without issuing real system commands.

To this end, we introduce NeuralOS, a first step toward realizing this vision. NeuralOS simulates an operating system's graphical interface entirely using deep neural networks. By modeling the OS interface as a generative process, it directly predicts graphical frames from user input events, such as mouse movements, clicks, and keyboard interactions, without manually programmed kernels or applications. Figure 1 illustrates an example sequence generated by NeuralOS, demonstrating realistic cursor movements and window interactions predicted solely from user inputs.

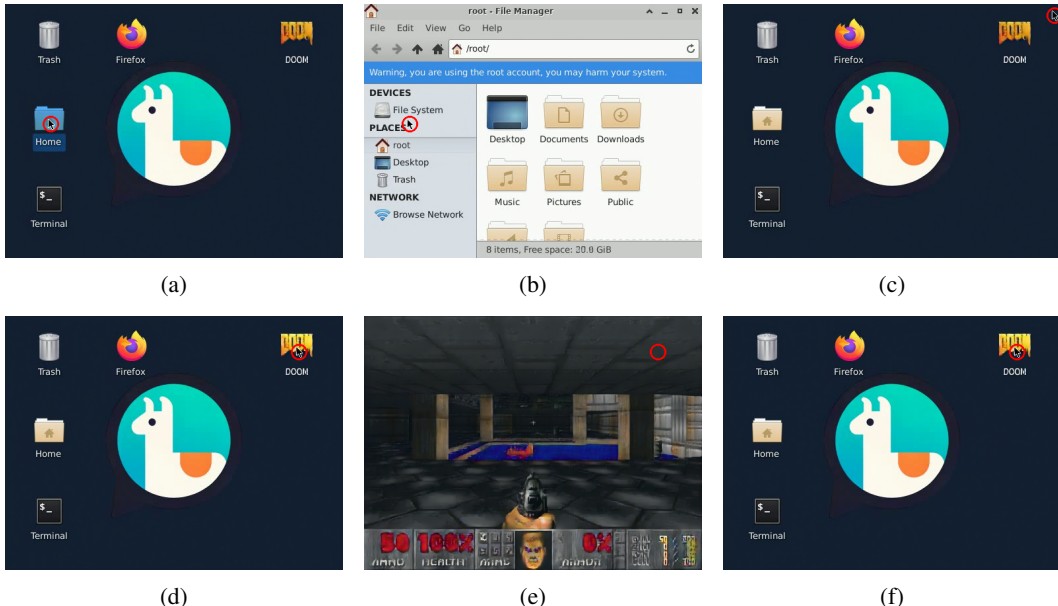

Figure 1: Real image sequence predicted by NeuralOS, illustrating the model's ability to simulate realistic GUI interactions. The sequence shows key frames as a user (a–c) opens and closes the "Home" folder, followed by (d–f) launches and closes a Doom application that was trained into the model using *synthetic* demonstrations. Cursor positions are highlighted with red circles. Frames are generated autoregressively, conditioned on previous frames and user inputs.

NeuralOS integrates two complementary architectures, analogous to the separation between OS kernels and desktop rendering programs: a recurrent neural network (Hochreiter & Schmidhuber, 1997) maintains internal computer states (such as open applications, hidden windows, and recent actions), while a diffusion-based convolutional neural renderer generates screen images. We train NeuralOS end-to-end on interaction sequences recorded from Ubuntu XFCE environments, combining randomly generated and realistic AI-generated human-like interactions.

Developing NeuralOS posed several challenges. (1) Long-term state tracking was essential due to delayed interface responses (e.g., opening Firefox could take up to 30 frames); we addressed this by using an RNN-based state representation. (2) Precise cursor modeling required explicit positional encodings within our diffusion model. (3) Without pretrained encoders for GUI interactions, we developed a novel pretraining method in which the RNN outputs were pretrained via regression losses and subsequently integrated into the diffusion model via finetuning. (4) Exposure bias during inference was mitigated using scheduled sampling techniques (Bengio et al., 2015; Deng et al., 2023; Ranzato et al., 2015). (5) Extensive engineering was necessary for scalable data collection and real-time inference, leveraging parallel Docker environments and AI-generated user interactions.

Experiments show that NeuralOS can generate realistic screen sequences, accurately model mouse interactions, and reliably simulate transitions such as application launches. While computational constraints limit its ability to precisely model fine-grained keyboard inputs, NeuralOS represents a step toward neural operating systems that adapt interfaces in real time, potentially enabling users to interact through natural language or gestures rather than fixed menus.

Beyond mimicking an existing operating system, NeuralOS can in principle learn user interfaces from demonstrations even if they are artificially constructed and do not exist in reality. As a proof of concept, we created a Doom application by combining fabricated desktop interactions with Viz-Doom gameplay recordings. NeuralOS learned to launch, play, and close Doom despite the application never being installed in the underlying system. This illustrates the broader principle that synthetic demonstrations, once distilled into a generative model, become usable user interfaces.

Our code, pretrained models, and an interactive demo are at `https://neural-os.com`.

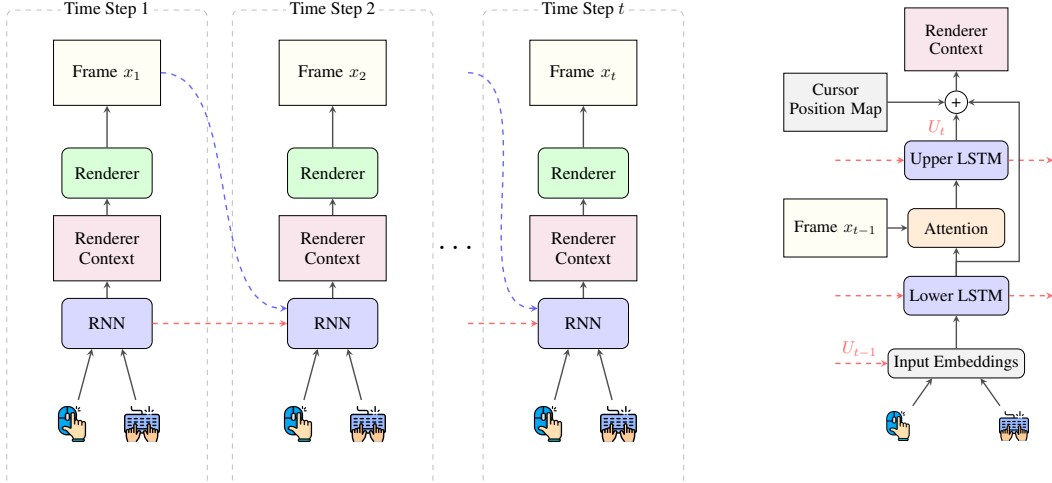

(a) High-level temporal architecture of NeuralOS.

(b) RNN structure at step $t$.

Figure 2: **NeuralOS Model Architecture.** (a) **High-level architecture of NeuralOS.** At each timestep, an RNN tracks the operating system's internal state based on user inputs (cursor positions, mouse clicks, keyboard events) and previously generated frames. This state is then passed as context to a diffusion-based renderer (UNet) that generates the next graphical frame. (b) **Detailed two-level RNN structure at timestep t.** The lower-level LSTM encodes user inputs, and then integrates visual information from the previous frame using attention. Its output is passed to the upper-level LSTM, which further processes these attention-informed representations. Feedback from the upper-level LSTM to the lower-level LSTM ($U_{t-1}$) ensures that the lower-level LSTM maintains awareness of upper-level state context and previous attention results. The combined outputs of both LSTMs, and cursor position encoding, form the renderer context. This hierarchical structure maintains constant computational complexity per timestep and supports continuous state updates during inference, essential for real-time OS interface simulation.

## 2 GENERATIVE MODELING OF OPERATING SYSTEM INTERFACES

We formalize the task of simulating operating system (OS) graphical interfaces as an autoregressive generative modeling problem. At each discrete timestep $t$, the model predicts the next graphical frame $x_t$ based on the sequence of previously observed frames $x_{<t} = x_0, x_1, \ldots, x_{t-1}$ and user input events $a_{\leq t} = a_1, a_2, \ldots, a_t$ up to and including the current timestep.

Formally, each frame $x_t$ is represented as an image tensor $x_t \in \mathbb{R}^{H \times W \times C}$, with $H$ and $W$ denoting image height and width, and $C$ the number of color or feature channels. The input event $a_t$ at timestep $t$ includes cursor coordinates $(x, y)$, binary indicators for mouse clicks (left or right), and a binary vector indicating which keyboard keys are pressed or released.

The probability distribution of an OS graphical sequence given user inputs can be expressed as:

$$P(x_{1:T} \mid a_{1:T}; \theta) = \prod_{t=1}^{T} P(x_t \mid x_{<t}, a_{\leq t}; \theta), \tag{1}$$

where $\theta$ represents the parameters of the neural generative model.

Unlike video generation (Singer et al., 2022b; Ho et al., 2022b; OpenAI, 2024), OS simulation must respond instantly to unpredictable user inputs, often causing abrupt changes in the interface, such as when a new application is launched. This contrasts with the smooth, predictable transitions typical in video generation. As a result, the model must maintain accurate and responsive state tracking. Next, we describe the NeuralOS architecture and training strategies designed for these requirements.

## 3 NEURALOS ARCHITECTURE

NeuralOS adopts a modular architecture inspired by the functional separation in traditional operating systems between kernel-level state management and graphical user interface (GUI) rendering. It comprises two primary components: a recurrent neural network (RNN) responsible for maintaining internal system states, and a diffusion-based renderer that generates graphical frames based on these states (see Figure 2a).

**Latent Diffusion Representation**  NeuralOS uses a latent diffusion framework (Rombach et al., 2022). We train an autoencoder to compress high-resolution OS screen images into lower-dimensional latent representations, reducing spatial dimensions by a scaling factor $s$. All modeling is performed within this latent space. At inference time, the generated latent frames are decoded back into pixel-level images only for users.

**Hierarchical RNN for State Tracking**  NeuralOS employs a hierarchical two-level RNN architecture to track the system state (see Figure 2b). Unlike transformers (Vaswani et al., 2017), whose inference complexity increases with context length, the RNN maintains constant complexity per timestep, which is crucial for continuous, long-horizon OS simulation. Existing video-game models (Feng et al., 2024; Oh et al., 2015; Valevski et al., 2024) typically rely on short context windows, which are sufficient because most game states are visually observable from recent frames. In contrast, the RNN state enables NeuralOS to recall previous interactions far in the past (Table 2).

At each timestep $t$, user inputs $a_t$ are encoded into embeddings. Specifically, cursor coordinates are discretized screen positions $(a_t^x, a_t^y)$, mouse clicks are binary indicators, and keyboard keys are binary press/release states. Each component is embedded separately and then concatenated:

$$\text{embed}(a_t) = \text{concat}(\text{embed}(a_t^x),\ \text{embed}(a_t^y),\ \text{embed}(a_t^{\text{L click}}) + \text{embed}(a_t^{\text{R click}}) + \sum_{\text{key}} \text{embed}(a_t^{\text{key}})).$$

These embeddings are processed by a lower-level LSTM, which also takes its previous hidden state $l_{t-1}$ and feedback from the previous upper-level LSTM state $U_{t-1}$ as inputs:

$$L_t, l_t = \text{LSTM}_{\text{lower}}(l_{t-1}, \text{concat}(\text{embed}(a_t), U_{t-1})),$$

where $l_t$ denotes the hidden state and $L_t$ denotes the corresponding output at timestep $t$.

To handle inherent uncertainties in OS behaviors, such as unpredictable application response times, the lower-level LSTM output $L_t$ is used as a query vector to attend over the previous graphical frame using multi-headed attention (Vaswani et al., 2017):

$$c_t = \text{MultiHeadedAttention}(\text{query} = W_q L_t, \text{keys/values} = W_k x_{t-1} + E_{\text{pos}}),$$

where $W_q, W_k$ are learnable projections and $E_{\text{pos}}$ encodes positional information of the latent frame.

The attention output $c_t$ is then merged with the original lower-level LSTM output $L_t$:

$$C_t = L_t + W_o c_t,$$

then processed by the upper-level LSTM:

$$U_t, u_t = \text{LSTM}_{\text{upper}}(u_{t-1}, C_t).$$

To ensure that the lower-level LSTM maintains awareness of higher-level contexts, the upper-level LSTM's output $U_t$ is fed back as an input to the lower-level LSTM in the next timestep.

**Spatial Encoding of Cursor Positions**  Precise cursor localization is critical for realistic OS interactions. Interactive world and videogame models (Parker-Holder & Fruchter, 2025; Xiang et al., 2025; Alonso et al., 2024; Valevski et al., 2024) typically use a few actions to navigate the world. In contrast, the cursor represents a much larger discrete action space that grows in screen size. This made cursor rendering extremely challenging, where errors in cursor location were often hundreds of pixels as shown in Figure 4b. To counter this, NeuralOS explicitly encodes cursor positions using a Gaussian spatial map $M_t \in \mathbb{R}^{H \times W}$. Instead of using a one-hot cursor position (which can lose

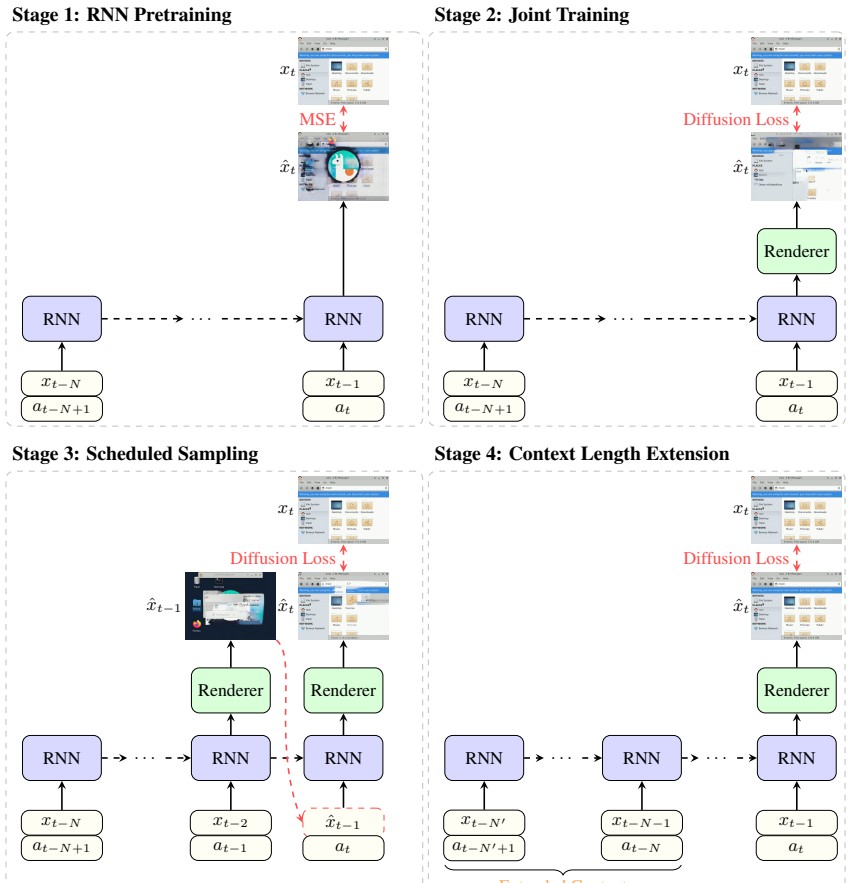

Figure 3: **Multi-stage training pipeline for NeuralOS.** (1) **RNN Pretraining:** The RNN is pretrained to predict latent frames using a mean squared error (MSE) loss. (2) **Joint Training:** The pretrained RNN and the diffusion-based renderer are jointly optimized using a standard diffusion loss. (3) **Scheduled Sampling:** To mitigate error accumulation caused by exposure bias, the most recent input frame is occasionally replaced by a previously generated frame. (4) **Context Length Extension:** The input context is extended to enable the model to capture long-term dependencies.

precision due to latent resolution constraints), we construct a Gaussian map centered at the cursor's scaled coordinates:

$$M_t(i,j) = \exp\left(-\frac{(i - a_t^x/s)^2 + (j - a_t^y/s)^2}{2}\right).$$

As demonstrated in Figure 4b, this spatial encoding is vital for accurate cursor rendering. This map $M_t$, combined with LSTM outputs $L_t, U_t$, forms the renderer context $R_t \in \mathbb{R}^{H \times W \times C'}$:

$$R_t = \text{concat}(W_L L_t, W_U U_t, M_t).$$

**Diffusion-Based Renderer**  To render the screen image, a UNet-based diffusion renderer generates the latent graphical frames conditioned on the renderer context $R_t$ (Ronneberger et al., 2015).

$$x_t \sim P_\theta(\cdot \mid R_t).$$

We concatenate the noisy image with $R_t$ as input to the UNet, and then predict the clean image.

## 4  MULTI-STAGE TRAINING APPROACH

Training NeuralOS is challenging due to ineffective use of RNN outputs, error accumulation during inference, and difficulties in capturing long-term dependencies due to computational constraints. To address these challenges, we take a multi-stage training approach (Figure 3).

**Stage 1: RNN Pretraining**  Unlike text-to-image diffusion models (Rombach et al., 2022), which use pretrained textual encoders, NeuralOS uses a customized RNN without pretrained checkpoints. Our initial experiments show that direct joint training leads to the renderer ignoring RNN outputs, as indicated by negligible gradient flow into the RNN. The diffusion-based renderer receives two streams of inputs: noisy latent frames and the RNN output; and without proper initialization of the RNN, it relies solely on the noisy image inputs, resulting in an under-trained RNN.

To address this, we first pretrain the RNN. We structure the RNN output $R_t \in \mathbb{R}^{H \times W \times C'}$ to match the spatial dimensions of the latent frame $x_t \in \mathbb{R}^{H \times W \times C}$, but with more channels ($C' > C$). The RNN is pretrained to predict the latent frames $x_t$ using a mean squared error (MSE) loss:

$$\mathcal{L}_{\text{MSE}} = \|R_t[:, :, : C] - x_t\|_2^2.$$

After pretraining, the RNN-generated frames alone tend to be blurry due to averaging multiple plausible outcomes, but crucially provide a strong initialization for subsequent joint training.

**Stage 2: Joint Training**  We jointly optimize the pretrained RNN and the diffusion-based renderer with a standard diffusion loss. The meaningful latent representations learned in RNN pretraining enable the renderer to use the RNN outputs, thus preventing the RNN outputs from being ignored.

**Stage 3: Scheduled Sampling**  During inference, errors accumulate over time and progressively degrade the quality of the generated frames. This issue arises from exposure bias (Ranzato et al., 2015): a model trained exclusively on ground-truth frames becomes overly reliant on perfect inputs and struggles when forced to operate on its own imperfect predictions during inference.

To mitigate this, we introduce a scheduled sampling training stage (Ranzato et al., 2015; Deng et al., 2023): during training, we replace the most recent input frame $x_{t-1}$ with the model-generated frame $\hat{x}_{t-1}$ with a small probability $p$. This method makes the model robust against input noise, thus mitigating error accumulation and improving generation stability over extended interactions.

**Stage 4: Context Length Extension**  Although NeuralOS can model sequences of arbitrary length, hardware memory limits justify shorter sequences during training, which limits exposure to long-term dependencies. To address this, we introduce a final stage that extends training to longer contexts, following initial training on short context windows for efficiency.

**Curriculum Training on Challenging Transitions**  In our collected OS interaction dataset, a substantial portion of frame transitions involve minor variations, such as slight cursor movements, which exhibit limited learning signals. To prioritize learning of significant OS state changes (e.g., opening menus or launching applications), we first train NeuralOS exclusively on challenging transitions. Specifically, we define challenging transitions as those frames whose pixel differences exceed a specified threshold: $\|x_t - x_{t-1}\|_2^2 > \epsilon$. Subsequently, we expand training to the full dataset. We apply this curriculum strategy for Stage 1 (RNN Pretraining) and Stage 2 (Joint Training).

**Additional Finetuning with Real-User Data**  After deploying NeuralOS, we collect a set of real-user interaction demonstrations and find that finetuning the trained model on these real-world examples improves its performance on frequent user tasks. This adaptive finetuning is conducted continuously through an interactive training framework introduced by Zhang et al. (2025), enabling the model to dynamically incorporate real-time collected data and achieve improved alignment with actual user behavior. Full methodological details and results can be found in that work.

## 5  Data Collection

**Agent-Based Demonstrations**  To collect realistic user interactions, we use Anthropic's Claude-3.5-Sonnet computer-use agent (Anthropic, 2024), which processes screenshots and invokes provided interaction functions. To maximize interaction diversity, we structure the agent's exploration process around a state-space search tree representing various OS states (see Figure 13 in Section J).

Specifically, we prompt the agent to identify all interactable GUI elements by moving the cursor to each element's center and reporting its bounding box (center, width, height). Each identified GUI element becomes a node in a search tree rooted at the initial OS state. The agent is then guided

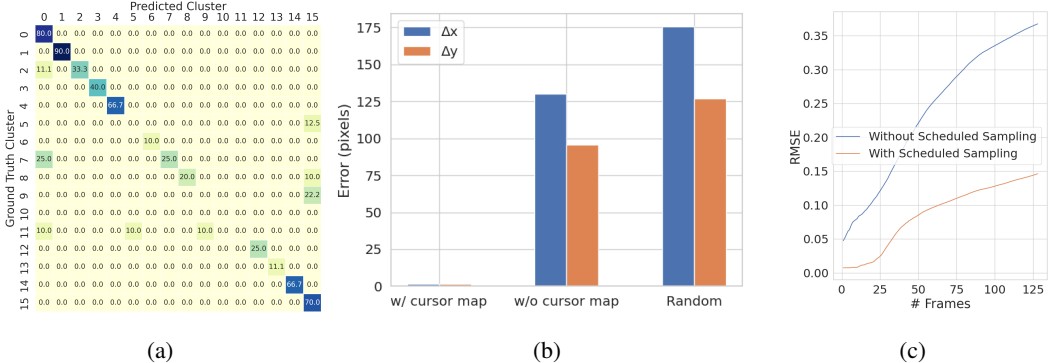

(a)                                    (b)                                    (c)

Figure 4: (a) Heatmap illustrating predicted vs. ground truth state transitions. Each cell represents the percentage of predictions assigned to a particular predicted cluster (x-axis), given a ground-truth cluster (y-axis). Only the top 16 clusters are displayed here; refer to Figure 11 for the complete heatmap. (b) Comparison of cursor position errors for NeuralOS (with cursor position map), NeuralOS without the cursor position map, and a random baseline. (c) Pixel RMSE of generated frames of using versus not using stage 3 scheduled sampling training.

through this tree: for each node, it moves the cursor to the corresponding GUI element and performs single or double clicks to transition to a new OS state, which then becomes a child node in the tree. We iteratively expand the tree until reaching a predefined maximum depth.

Next, we initiate further interactions from each leaf node, allowing the agent to explore freely from these distinct OS states for a fixed duration, thereby capturing diverse interaction sequences.

**Random Exploration**    We find that relying exclusively on agent-generated demonstrations introduces spurious correlations. For example, the model incorrectly associates cursor movement toward a window's close button with the action of closing, even in the absence of a click. To mitigate such unintended associations, we supplement the dataset with random interaction data.

In generating these random interactions, we simulate mouse movements, clicks, and keyboard inputs (key presses and releases) stochastically. To improve realism, we introduce several constraints and heuristics iteratively developed through experimentation. Cursor movements are modeled using Bezier curves to emulate natural mouse trajectories (Mortenson, 1999). Double-click events, rare under purely random sampling, are explicitly generated. Additionally, we enforce constraints such as limiting simultaneous key presses and ensuring keys are released only if previously pressed.

## 6 EXPERIMENTAL SETUP

**Data**    We collected data using 64 parallel Docker containers, each with Ubuntu 20.04 and XFCE desktops at a resolution of $512 \times 384$. To simplify the environment, the desktop was limited to four applications: *Home*, *Trash*, *Terminal*, and *Firefox*. Data consisted of 2K agent-based and 120K random exploration demonstrations, each 30 seconds long at 15 fps, resulting in 12TB of latent data after compression via an autoencoder. The autoencoder reduced the images by a factor of 8 to a latent resolution of $64 \times 48$ with 16 channels. Details of the autoencoder are in Section I.

**Model**    The hierarchical RNN has two LSTM modules (each with hidden size 4,096) and a multi-headed attention module (8 heads, 1,024 total dimension). The RNN output is projected to 32 channels and concatenated with the noisy latent frame (16 channels). The UNet uses four resolution levels with channel multipliers of [1, 2, 3, 5], two residual blocks per level, and attention layers at resolutions 8, 4, and 2. It has a base model dimension of 192 and outputs 16 channels. The final model contains 2.2B parameters for the RNN and 263M parameters for the UNet renderer.

**Training and Inference**    NeuralOS was trained using our proposed multi-stage training approach. See Section C for full details on hyperparameters. The total data processing and training took

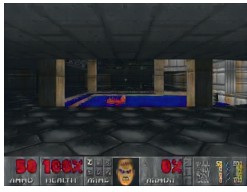 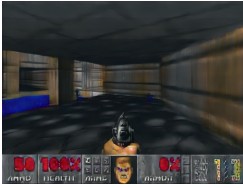 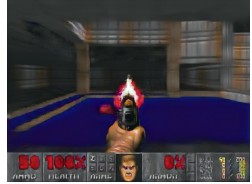 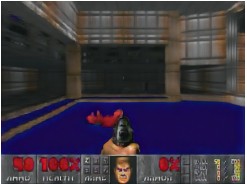

Figure 5: Doom interactions generated by NeuralOS. Doom was never installed in the underlying operating system used for data collection; instead, the model learned to simulate the application from synthesized training data, producing realistic walking and shooting behavior from user inputs.

Table 1: Human success rate (%) of identifying the real OS.

| Crop Setting | 10s | 20s | 30s | 40s | 50s | 60s |
|---|---|---|---|---|---|---|
| No crop | 58.3 | 55.0 | 60.0 | 56.7 | 61.7 | 59.3 |
| Bottom cropped (40px) | 51.7 | 53.3 | 50.0 | 48.3 | 46.7 | 50.0 |

approximately 4 months, requiring 17,000 GPU hours on a server with 8 NVIDIA H200 GPUs (141GB memory per GPU) and an additional 6,000 GPU hours on a server with 8 NVIDIA H100 GPUs (80GB memory per GPU). At inference time, we used DDIM sampling (Song et al., 2020a) with 2 steps, achieving an inference speed of 18 fps on a single NVIDIA H100 GPU.

# 7 EXPERIMENTS

Given the substantial computational resources required to train NeuralOS, our evaluation focused on NeuralOS variants, ablations, and intermediate training phases. For all evaluations, we used a subset of 730 examples from a reserved evaluation dataset, unless mentioned otherwise.[1]

**Cursor Position Modeling** We evaluated cursor-position accuracy by training a regression model to detect cursor coordinates from the generated images. With the cursor position map, NeuralOS achieved highly accurate cursor localization, with an average position error of $\Delta x = 1.6$ and $\Delta y = 1.4$ pixels (Figure 4b). Given the $512 \times 384$ resolution of the images, this corresponds to less than 0.5% of the frame width or height, indicating that cursor locations in generated images are precise. This performance significantly outperformed a baseline without cursor position maps[2] ($\Delta x = 130.0$, $\Delta y = 95.8$) and the random baseline ($\Delta x = 175.4$, $\Delta y = 126.9$), confirming the importance of explicit spatial encoding for accurate cursor localization.

**State-Transition Modeling** Most OS interactions involve only cursor movements without significant visual changes. To focus evaluation on the crucial moments where the interface undergoes meaningful changes, such as opening or closing an application, we identified challenging frame transitions from the evaluation set.[3] These transitions were clustered into 73 categories using DB-Scan. NeuralOS predictions were then matched against the nearest cluster labels and compared with the ground truth. As shown in Figure 4a, NeuralOS achieved an accuracy of 37.7% (correct state transitions over total state transitions) captured on the diagonal. This substantially outperformed majority voting (1.4%). Note that off-diagonal predictions may still correspond to valid outcomes due to inherent timing variability in OS actions (see Section D).

**Human Evaluation** We conducted a human evaluation to test how indistinguishable NeuralOS outputs are from a real operating system. Human raters were shown side-by-side video clips of the same user interaction sequence, one generated by NeuralOS and one recorded from an Ubuntu

---

[1]This number was chosen because our clustering procedure (detailed later) identified 73 clusters of challenging frame transitions, and we selected 10 examples per cluster, resulting in a total of 730 examples.

[2]This baseline is an earlier model version trained for 700K steps under slightly different conditions.

[3]These were those with mean pixel distance greater than 0.1 from the previous frame to the target frame.

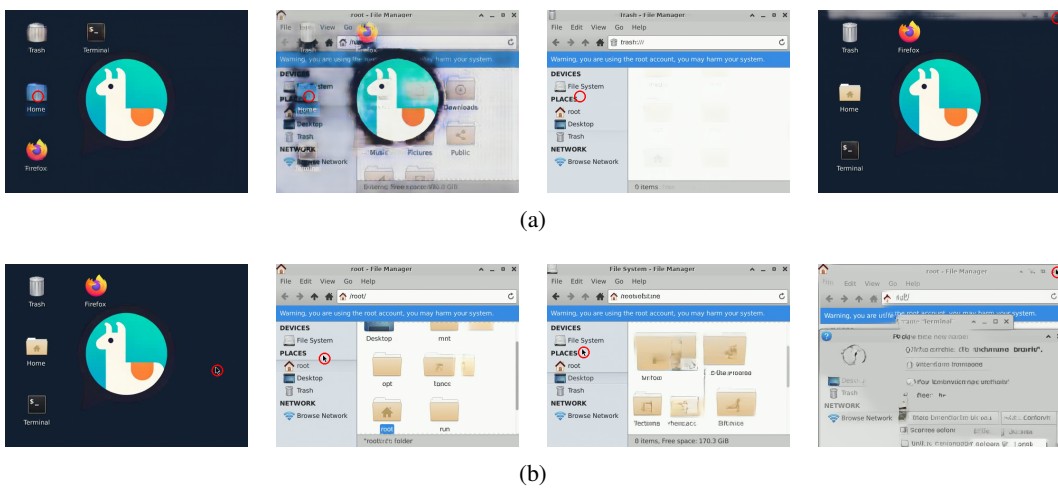

Figure 6: Ablation studies. (a) shows the limitations of directly using RNN outputs. (b) shows error accumulation without scheduled sampling, demonstrating its necessity for maintaining stability.

XFCE desktop, and asked to identify the real system. Each clip lasted 10–60 seconds (sampled at 10s intervals) and was drawn from real human interactions collected through our online demo.

To control for artifacts such as fluctuating disk-space counters (a known issue in diffusion), evaluation was performed under two conditions: (1) original videos and (2) videos with bottom 40 pixels (10% of height) cropped. In each condition, participants judged 30 randomly ordered comparisons.

As shown in Table 1, participants performed only slightly better than chance on up to 20 seconds of interactions, suggesting that NeuralOS is visually realistic for basic interactions.

**Learning from Synthetic Demonstrations** A benefit of an end-to-end learned user interface is its ability to learn from demonstrations that do not correspond to real implementations. To illustrate this, we constructed a Doom application entirely from synthetic data, even though Doom was never installed in the docker. Desktop screenshots were augmented with a Doom icon, synthetic double-click actions were inserted to simulate launching the game, VizDoom gameplay segments with corresponding input events were spliced in, and each sequence ended with an ESC action returning to the desktop. Trained on these demonstrations, NeuralOS learned to launch, play, and close Doom as if it was a native application (Figures 1 and 5).

**Long-Term Memory** We evaluated whether NeuralOS has long-term memory using a folder creation task. We navigated to *Home* → *Documents*, created a "New Folder" (via right-click → *Create Folder*), closed the file manager, and later reopened *Documents* after a delay. During training, the model saw only short delays: we included 500 examples where the file manager was reopened after 8 frames in the training set. At test time, we extended the delay to 64, 128, and 256 frames. As a control, we also included 500 examples where no folder was created.

Folder presence was evaluated by checking the region where the folder icon should appear, comparing it to the ground truth with an MSE threshold. Table 2 shows the model can mostly recall whether a folder had been created even after 256 frames, which is far beyond both the 8-frame training delay and its maximum training context of 64 frames. This indicates that NeuralOS develops persistent memory representations of application state that generalize well beyond its training horizon.

**Ablations** Without joint training (relying solely on the pretrained RNN), the predictions are blurry (Figure 6a). This is caused by the MSE loss encouraging the model to predict averaged representations of multiple plausible outcomes rather than committing to a single clear target. Additionally, cursor positions were absent, despite the model correctly capturing state transitions (e.g., opening the home folder), indicating that the RNN still implicitly encoded cursor information.

Table 2: Folder presence modeling confusion matrix after reopening file manager with increasing gaps. The higher the diagonal values (bolded), the better.

| Frame Gap | Ground Truth | Predicted | |
|---|---|---|---|
| | | Has Folder | No Folder |
| 64 | Folder Created | **0.52** | 0.48 |
| | No Folder Created | 0.03 | **0.97** |
| 128 | Folder Created | **0.60** | 0.40 |
| | No Folder Created | 0.00 | **1.00** |
| 256 | Folder Created | **0.62** | 0.38 |
| | No Folder Created | 0.02 | **0.98** |

Omitting scheduled sampling led to rapid deterioration in generated frame quality due to compounding prediction errors (Figure 6b). In contrast, incorporating scheduled sampling greatly improved the model's robustness (Figure 1), which is also evident from quantitative evaluation in Figure 4c.

## 8 CONCLUSION AND FUTURE WORK

We introduced NeuralOS, a system that simulates OS graphical interfaces using generative models. Trained end-to-end on interaction sequences, NeuralOS produces realistic screen sequences, accurately predicts mouse interactions, and captures transitions such as opening applications.

A benefit of end-to-end trained interfaces is their ability to learn from *synthetic* demonstrations. Our Doom experiment illustrated this: the model simulated launching, playing, and closing Doom, even though the application was never installed in the underlying operating system. This points to a broader paradigm for future generative interfaces: if consistent demonstrations can be provided, even if they are manually edited, stylized, or fabricated, they can be internalized into usable interfaces.

## REPRODUCIBILITY STATEMENT

Our training code and data processing scripts are available at `https://github.com/yuntian-group/neural-os`. An interactive demo is hosted at `https://neural-os.com/`, with demo code at `https://github.com/yuntian-group/neuralos-demo`. The model checkpoints used by the demo are publicly available on Hugging Face (linked directly from the demo repository). Additional details on training procedures and dataset construction are provided in the Appendix.

## ETHICS STATEMENT

Generative models capable of imitating software inevitably raise broader considerations around intellectual property, licensing, and the economics of software distribution, similar to the concerns that emerged when large (vision) language models began reproducing copyrighted text, API patterns, or proprietary knowledge (He et al., 2025b; Wei et al., 2024; Cooper et al., 2025). To avoid these issues, training in this work was conducted on Ubuntu XFCE environments. More broadly, this project should be understood as an early research prototype investigating the feasibility of generative user interfaces rather than a system intended to replicate or redistribute commercial applications.

## ACKNOWLEDGMENTS

This research was supported by Compute Canada through the Resources for Research Groups 2025 competition, awarded to Yuntian Deng (RRG No. 5275), and was also partially supported by collaborative research funding from the National Research Council of Canada's Artificial Intelligence for Design Program (AI4D-150). Additionally, Yuntian Deng acknowledges support from an NSERC Discovery Grant (RGPIN-2024-05178) and a Starter Grant provided by the University of Waterloo.

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

## A   BASELINE COMPARISON WITH DIAMOND

We adapted the DIAMOND model (Alonso et al., 2024), which was developed for action-conditioned video game simulation (e.g., Atari and CS:GO), to the operating-system setting. DIAMOND serves as a representative action-conditioned diffusion model capable of predicting future frames given past frames and actions.

**Adaptation**   DIAMOND was designed for video game environments with short-horizon temporal dependencies and coarse action spaces. To ensure a fair comparison, we modified it in the following ways:

- **Context window size**: Increased from $4 \rightarrow 64$ past frames/actions to accommodate longer temporal dependencies in desktop interactions (e.g., launching Firefox may take 20–30 frames).

- **Cursor coordinate discretization**: Expanded from the original $23 \times 17$ grid to match the $512 \times 384$ desktop resolution. This supports pixel-accurate pointing (e.g., clicking the close icon).

- **Keyboard action space**: Expanded from the original 11 keys to the full keyboard (179 keys).

- **Training data**: Trained on the same dataset used for NeuralOS to enable comparison.

Apart from these modifications, we followed the original architecture and training setup as closely as possible. Both DIAMOND and NeuralOS were trained from scratch under matched compute constraints (single B200 GPU for around 48 hours). NeuralOS in the main experiments was later trained longer (around 3M steps), but here we report results under equalized compute.

**Qualitative Results**   Figure 7 illustrates generations from the three models evaluated under the same interaction sequence used in the main paper (opening the "Home" folder, closing it, launching Doom, and returning to the desktop). The adapted DIAMOND model produces decent frames before the first abrupt transition occurs (opening Home), but generation quality collapses afterwards.

To improve the stability and reduce accumulated error over time, we also fine-tuned DIAMOND with scheduled sampling ($p = 0.05$) for an additional 50k steps. This variant could recover from degenerations, but it still failed to execute application-launch transitions.

In contrast, the NeuralOS model produced more stable generations and was able to successfully open and close Doom, although at this training stage it had not yet learned to open the Home folder. (From the main experiments, we know that with longer training the model eventually learns this behavior.) Interestingly, the cursor is not rendered in these generations (we observed that learning to draw the cursor is particularly challenging because it occupies only a tiny fraction of the pixels and requires many gradient steps before it begins to appear). However, the model still responds to mouse hovers and double-clicks, indicating that the cursor position is being implicitly tracked even though its visual rendering has not yet emerged.

**Quantitative Results**   We additionally computed pixel RMSE between generated and ground-truth frames as a function of the number of generated steps. As shown in Figure 7, error for both DIAMOND variants increases rapidly, particularly after state-changing events such as opening an application. In contrast, NeuralOS maintains substantially lower error over longer horizons.

This comparison is not meant to imply that DIAMOND is unsuitable; rather, it highlights differences in modeling assumptions. Action-conditioned diffusion models developed for video games generally assume that the necessary state is visually encoded in recent frames, making short context windows sufficient. In operating-system settings, however, critical state may persist far beyond a short temporal window (e.g., whether a folder was created earlier in the workflow). With a fixed 64-frame context, DIAMOND cannot represent such long-term state by design. NeuralOS, through its recurrent state component, generalizes beyond its training horizon, consistent with the results in Table 2 showing correct folder-state recall after delays of up to 256 frames.

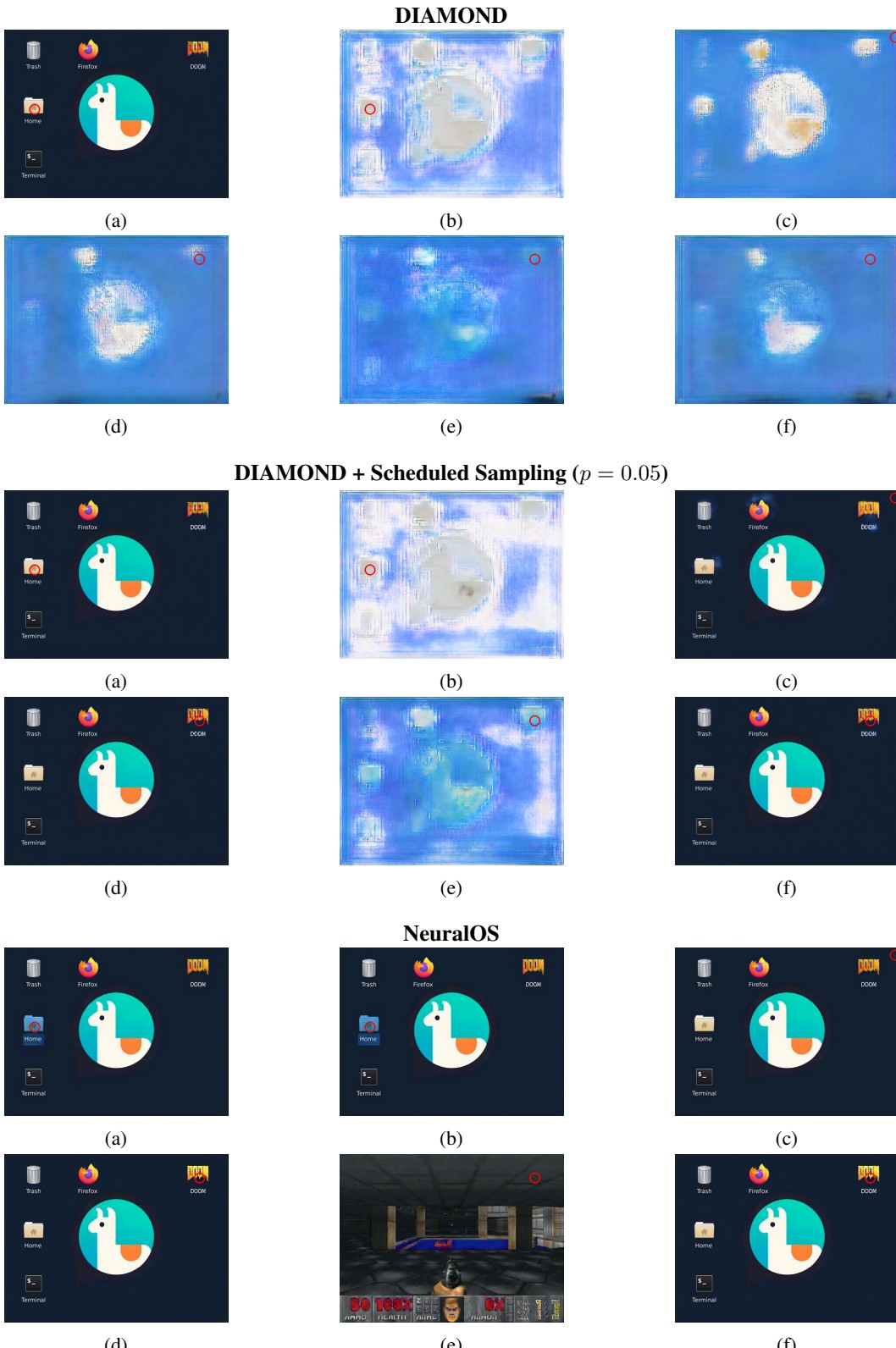

Figure 7: Generations from DIAMOND, DIAMOND with scheduled sampling, and NeuralOS evaluated on the same interaction trajectory used in Figure 1. Each pair of rows shows key frames as the model (a–c) opens and closes "Home", followed by (d–f) launching and closing Doom.

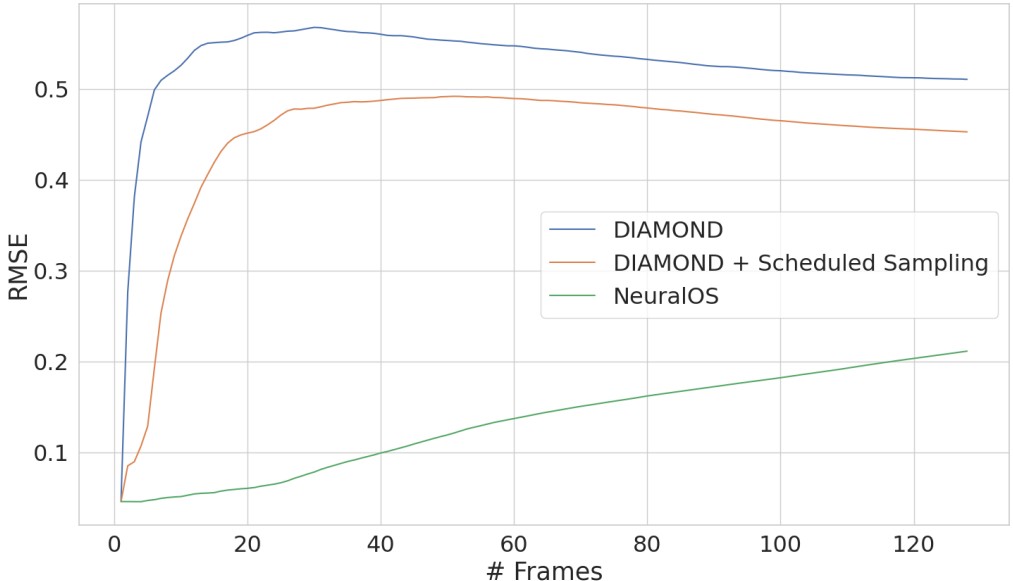

Figure 8: Pixel RMSE of generated frames of NeuralOS, DIAMOND, and DIAMOND with scheduled sampling. Both DIAMOND variants show increasing error over frames, particularly after state transitions such as application launch events. NeuralOS maintains lower error across longer horizons, reflecting better state tracking enabled by its recurrent architecture.

Table 3: Detailed hyperparameters and dataset configurations used for each stage of NeuralOS's multi-stage training. "Challenging transitions" are where the target frame differs from the preceding input frame by a mean pixel difference greater than 0.1. These challenging transitions constitute approximately 2.8% of the full dataset.

| Stage | Dataset | Batch | Steps | LR | Context | Sampling $p$ |
|---|---|---|---|---|---|---|
| *Stage 1: RNN Pretraining* | | | | | | |
|    Challenging transitions | 2.8% subset | 256 | 50K | $8 \times 10^{-5}$ | 32 | — |
|    Full dataset | 100% | 256 | 200K | $8 \times 10^{-5}$ | 32 | — |
| *Stage 2: Joint Training (RNN + Renderer)* | | | | | | |
|    Challenging transitions | 2.8% subset | 64 | 100K | $8 \times 10^{-5}$ | 32 | — |
|    Full dataset | 100% | 64 | 1M | $8 \times 10^{-5}$ | 32 | — |
| *Stage 3: Scheduled Sampling* | | | | | | |
|    Full dataset | 100% | 256 | 500K | $8 \times 10^{-5}$ | 32 | 0.05 |
|    Full dataset (lr reduced) | 100% | 256 | 500K | $2 \times 10^{-5}$ | 32 | 0.05 |
| *Stage 4: Context Length Extension* | | | | | | |
|    Full dataset | 100% | 128 | 100K | $2 \times 10^{-5}$ | 64 | 0.05 |

## B  LIMITATIONS

The current study uses Ubuntu XFCE as the operating environment. This choice reflects practical rather than technical constraints: Ubuntu provides a legally redistributable platform aligned with our goal of releasing all resources openly, and training comparable models across multiple operating systems would substantially increase data collection, storage, and compute requirements. Extending NeuralOS to more operating systems such as Windows or macOS remains an important direction for future work.

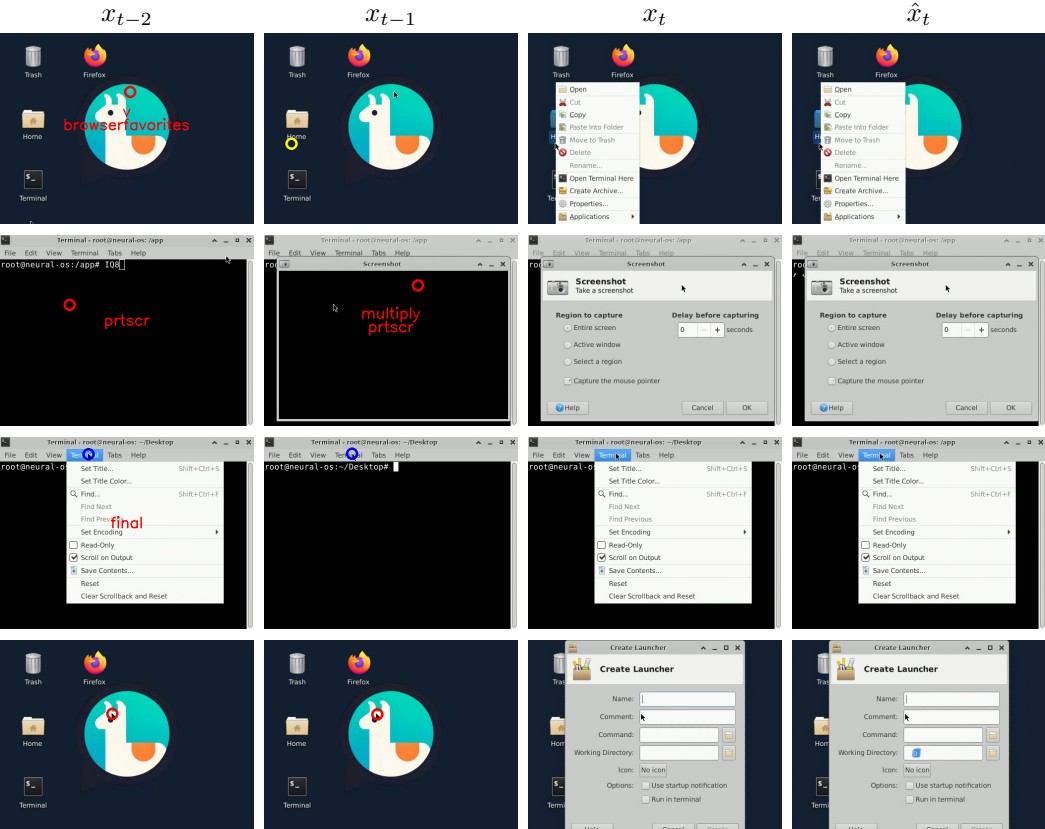

Figure 9: Correct prediction examples by NeuralOS. Each row shows two past frames (columns 1–2), ground-truth next frame (column 3), and NeuralOS's prediction (column 4). Cursor positions are marked one frame in advance with circles (red: move-only, blue: left-click, yellow: right-click). NeuralOS correctly captures various GUI transitions, including opening menus and launching applications.

Our work represents an initial step toward a fully generative OS, but several limitations remain. Despite substantial training compute (17,000 H200 GPU hours and 6,000 H100 GPU hours), NeuralOS is still far from replicating the capabilities of a real operating system: screen resolution remains low, fine-grained keyboard interactions are not reliably supported (Section D). For inference, the current system requires an H100 GPU and is not yet suitable for cost-efficient or practical deployment. This reflects the exploratory nature of the work. We expect future architectures, training strategies, and hardware to make such systems substantially more efficient. Additionally, many open challenges remain, including enabling the generative OS to interact with external resources (e.g., internet), and introduce controllability beyond traditional OS boundaries.

## C MULTI-STAGE TRAINING HYPERPARAMETERS

NeuralOS is trained in four stages. A conceptual overview of the role and outcome of each stage is shown in Table 4, and the corresponding hyperparameter settings and dataset configurations are detailed in Table 3.

In Stage 1 (RNN Pretraining), the RNN was trained first on the subset of challenging transitions, defined as examples whose target frame differs from the last input frame by an average pixel difference greater than 0.1. These challenging transitions constitute about 2.8% of the entire dataset. We used a batch size of 256 and an initial learning rate of $8 \times 10^{-5}$, training for 50K steps. Afterwards, the model was trained on the full dataset (100% of data) for an additional 200K steps, maintaining

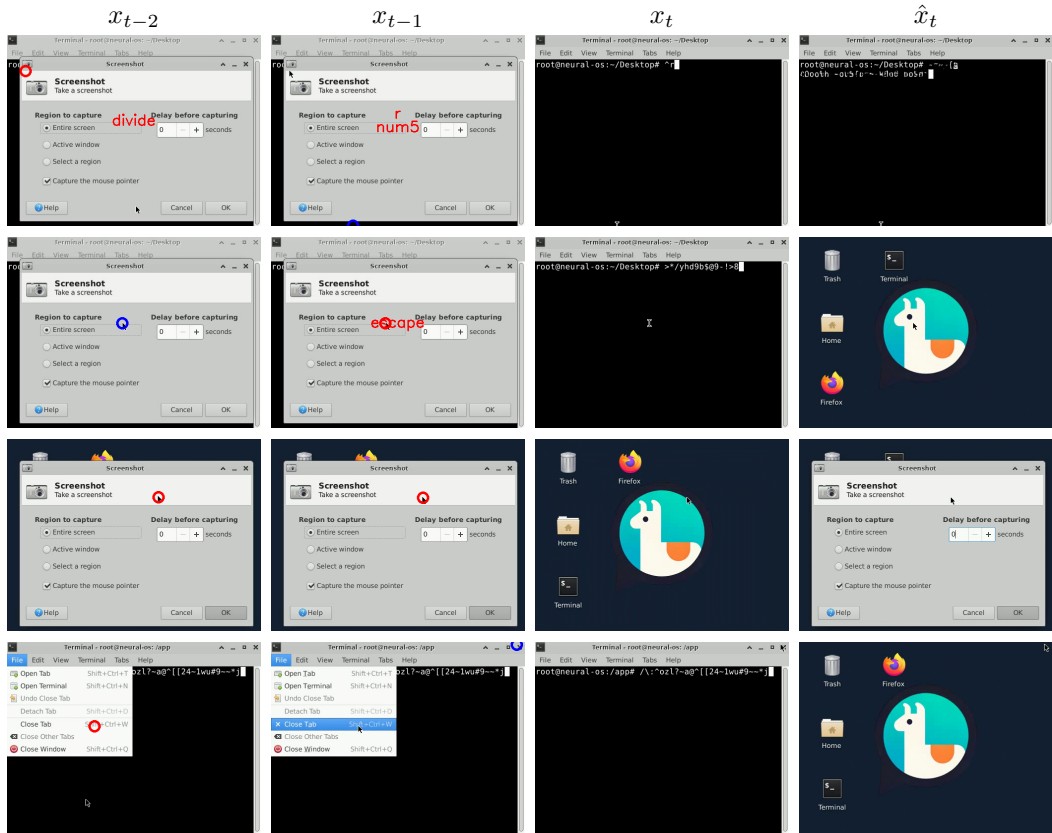

Figure 10: Prediction examples where the generated frame does not match the ground truth frame. The layout follows Figure 9. Note that not all mismatches represent errors. For example, the third row illustrates a case where the screenshot tool window is closed in the ground truth frame but remains open in the prediction. This discrepancy arises because the window-closing action (not shown due to the limited context window) can have variable timing. Thus, both the predicted and ground truth frames are valid outcomes in such scenarios.

the same batch size and learning rate. The context window length was fixed at 32 frames during this stage.

In Stage 2 (Joint Training), the pretrained RNN and the renderer were jointly trained end-to-end, first focusing on the challenging transitions (2.8% subset) for 100K steps, then extended to the full dataset for an additional 1M steps. The learning rate remained at $8 \times 10^{-5}$, with a reduced batch size of 64 to stabilize diffusion training. The context length remained 32 frames, and no scheduled sampling was applied in this stage.

In Stage 3 (Scheduled Sampling), we trained on the full dataset using scheduled sampling with probability $p = 0.05$, where the most recent past frame was occasionally replaced by a model-generated frame during training. Initially, training was conducted for 500K steps at a batch size of 256 and a learning rate of $8 \times 10^{-5}$. The learning rate was subsequently reduced to $2 \times 10^{-5}$, followed by an additional 500K training steps. The context window remained 32 frames.

Finally, in Stage 4 (Context Length Extension), we increased the context window length from 32 to 64 frames to enable NeuralOS to better capture long-term dependencies. Scheduled sampling probability was maintained at $p = 0.05$. We used a lower learning rate of $2 \times 10^{-5}$, reduced the batch size to 128 to fit GPU memory constraints, and finetuned the model for 100K additional steps.

Table 4: Summary of the four-stage training pipeline. Each stage addresses a specific modeling challenge.

| Stage | Purpose | Outcome | Remaining limitations |
|---|---|---|---|
| Stage 1 – RNN Pretraining | Train the RNN to predict latent frame embeddings so that it encodes cursor motion and keyboard inputs, ensuring the RNN is not ignored during later joint training. | Model learns basic transitions; decoding the RNN output via the autoencoder yields plausible but blurry frames. | Generations are blurry due to the MSE objective. |
| Stage 2 – Joint Training | Combine the pretrained RNN with the diffusion-based renderer so that the decoder can sharpen predictions while retaining temporal structure from the RNN. | Produces stable short-horizon interactions. | Exposure bias: during training the model only sees ground truth frames, but at inference it must condition on its own imperfect outputs, leading to compounding errors over time. |
| Stage 3 – Scheduled Sampling | Introduce scheduled sampling so the model is exposed to its own generated frames during training. | Improves stability over extended interaction sequences. | Context window length is still limited for training efficiency, and the model has not yet observed very long interaction horizons. |
| Stage 4 – Context Length Extension | Expand the temporal context so the model can learn longer interaction dependencies. | Enables the model to observe and learn longer-term dependencies. | Future work is needed on efficiency, controllability, and external tool integration. |

# D  QUALITATIVE ANALYSIS

We further analyze NeuralOS qualitatively by examining successful and unsuccessful generation examples, shown in Figure 9 and Figure 10, respectively. Each row illustrates two past frames, the ground-truth next frame, and NeuralOS's predicted frame. Cursor positions in past frames are annotated with colored circles to indicate the cursor's intended position at the next frame: red represents cursor movement only, blue denotes left-click actions, and yellow signifies right-click actions. Additionally, keys pressed at each frame are displayed in red text. Note that cursor annotations are shifted forward by one frame to clearly depict cursor positions expected in the immediate subsequent frame.

In Figure 9, NeuralOS accurately predicts various critical GUI transitions, such as launching applications and opening menus through both mouse clicks and keyboard inputs, demonstrating its ability to capture spatial and functional dynamics.

However, as shown in Figure 10, NeuralOS exhibits limitations, particularly for subtle actions like moving the cursor to a "Close Tab" button without clicking. Moreover, NeuralOS currently struggles to accurately represent fine-grained keyboard inputs, such as specific characters typed in a terminal.

It is worth noting that not all mismatches between predictions and ground truth constitute errors; some discrepancies arise from variable timing in GUI responses, exemplified in Figure 10.

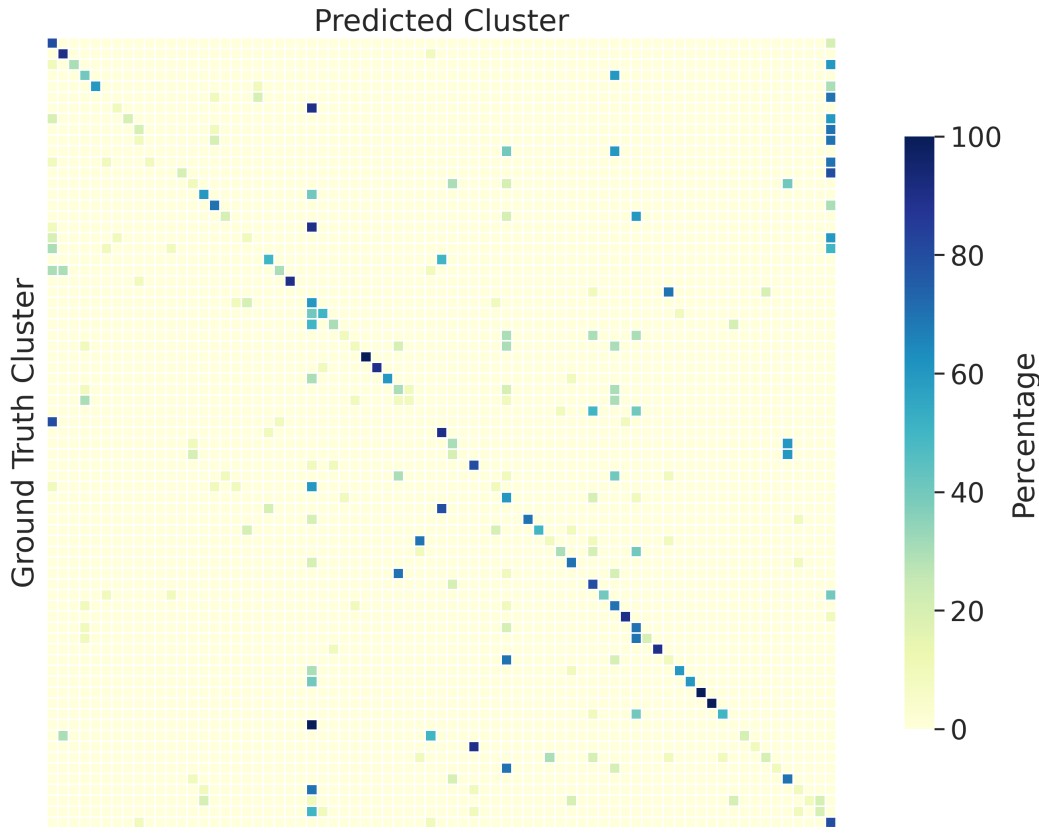

Figure 11: Complete heatmap of NeuralOS state transitions. Each cell represents the percentage of predictions corresponding to a predicted cluster (x-axis) given a ground-truth cluster (y-axis). Diagonal entries indicate exact cluster matches.

## E    FULL STATE TRANSITION HEATMAP

Due to space constraints, the main text presents only a truncated version of the state transition heatmap. In Figure 11, we provide the complete heatmap, showing NeuralOS's predictions across all identified clusters.

## F    INTERACTIVE WEB DEMO

To make NeuralOS accessible, we built an interactive web demo available at `https://neural-os.com`. The system consists of a browser-based frontend and a GPU-backed backend.

**Frontend**    The frontend renders the NeuralOS screen inside a single HTML canvas. Mouse movements, clicks, and keyboard inputs are streamed to the backend over a persistent WebSocket connection.

Because user input often arrives faster than the model can produce frames (about 55 ms per frame), the system performs lightweight input coalescing: redundant mouse-move events are dropped when a new meaningful event (e.g., click or keyboard press) occurs. This makes interaction feel smoother.

Some OS actions require multiple generated frames to complete (for example, launching Firefox may take about 20 frames). To handle these cases, the demo includes an "Auto Input" mechanism: when the user stops interacting, the frontend periodically resends the last cursor position (every 67 ms) so that the model continues progressing through delayed transitions rather than waiting for new input.

**Backend**    The backend uses a dispatcher–worker architecture. The dispatcher manages queueing, session lifetime, WebSocket routing, and worker availability. Each worker process runs one instance of NeuralOS and occupies a single GPU. Only one user can actively interact per worker.

When users are waiting, the system applies time-limited sessions to ensure fair GPU access. The demo also includes an idle timeout, which triggers a "Connection Timeout" warning before disconnecting users who stop interacting.

To reduce bandwidth and improve responsiveness, generated images are split into $32 \times 32$ pixel blocks, and only blocks that change meaningfully from the previous frame are transmitted to the browser.

**Debug Controls**    The demo includes internal debugging options such as toggling "Use RNN" (bypassing diffusion), changing sampling steps, resetting state, and modifying Auto Input behavior. These features were primarily used during development and are now hidden by default; advanced users can still access them via the browser console using `showDebugControls()`.

Notably, "Use RNN" mode decodes the RNN state directly, which is only valid after the RNN pretraining stage. After joint training with the diffusion model, this mode produces degraded frames because it is no longer the intended decoding path. The default configuration (full RNN + diffusion model) corresponds to the model used in the paper.

## G    TRANSFORMER VS RNN ENCODER

We initially implemented our system using a transformer-based encoder architecture. However, in interactive OS simulation, inference sequences can be arbitrarily long, as each user action generates a new frame that is appended to the model's context unless explicitly truncated. For long or continuous interactions, this results in a steady increase in transformer memory consumption during inference, ultimately leading to out-of-memory (OOM) failures for sufficiently long sequences.

To address this, we replaced the transformer with a hierarchical recurrent neural network (RNN) architecture. RNNs maintain a constant-size hidden state between steps, enabling inference-time GPU memory usage that is independent of sequence length. Table 5 shows the memory requirements of a RNN encoder vs a Transformer encoder. Moreover, computer state modeling is inherently Markovian, making the RNN a suitable choice.

Table 5: Inference-time GPU memory usage (GB) for transformer and RNN architectures of similar parameter counts. The transformer encoder cannot accommodate 1024 frames on an L40 GPU.

| Video Length (s) | Transformer (GB) | RNN (GB) |
|---|---|---|
| 4 | 21.61 | 11.96 |
| 64 | 22.69 | 11.96 |
| 128 | 23.81 | 11.96 |
| 256 | 26.07 | 11.96 |
| 512 | 30.60 | 11.96 |
| 1024 | >40 (OOM) | 11.96 |

Table 6: Application open–close accuracy. We report RMSE and mean absolute error (L1) between NeuralOS predictions and ground-truth frames, with pixel values normalized to [0, 1].

| Application | Action | RMSE | L1 |
|---|---|---|---|
| Terminal | Open | 0.021 | 0.001 |
| Firefox | Open | 0.048 | 0.006 |
| Trash | Open | 0.017 | 0.001 |
| Home | Open | 0.019 | 0.001 |
| Terminal | Close | 0.015 | 0.001 |
| Firefox | Close | 0.016 | 0.001 |
| Trash | Close | 0.016 | 0.001 |
| Home | Close | 0.016 | 0.001 |

# H  APPLICATION OPEN–CLOSE TRANSITIONS

We evaluated the accuracy of application open–close transitions in NeuralOS. Specifically, we constructed 150 sequences for four main applications (Terminal, Firefox, Trash, Home) by sampling random cursor paths and click locations uniformly within the icon bounding boxes.

We compared the predicted and ground-truth frames at the end of each open and close event using root mean squared error (RMSE) and mean absolute error (L1). To account for rendering latency, we evaluated the frame after a fixed delay following each click: 40 frames after Firefox double-clicks, 10 frames after other application double-clicks, and 5 frames after single-click closing. These delays were determined empirically to align with the frame at which the application's visual state change completes.

As shown in Table 6, the low RMSE and L1 scores confirm that NeuralOS accurately performs application open and close transitions, consistent with observations from the interactive demo.

# I  AUTOENCODER DETAILS

We trained a Variational Autoencoder to compress high-dimensional OS screenshots into low-dimensional latent representations suitable for efficient training of NeuralOS.

**Model Architecture**   The architecture of the autoencoder is based on the model proposed by (Rombach et al., 2022) with some custom adjustments to improve reconstruction quality. The encoder consisted of four convolutional downsampling blocks with 128 base channels. Each downsampling stage contained two residual blocks and no attention layers. The latent channel dimension was set to 16.

**Training**   The autoencoder was trained using a combined reconstruction and adversarial loss function. We trained the autoencoder using the Adam optimizer with a learning rate of 1e-6, a batch size of 10, and a total of 2 million training steps on our dataset. Training was conducted on a single NVIDIA H200 GPU.

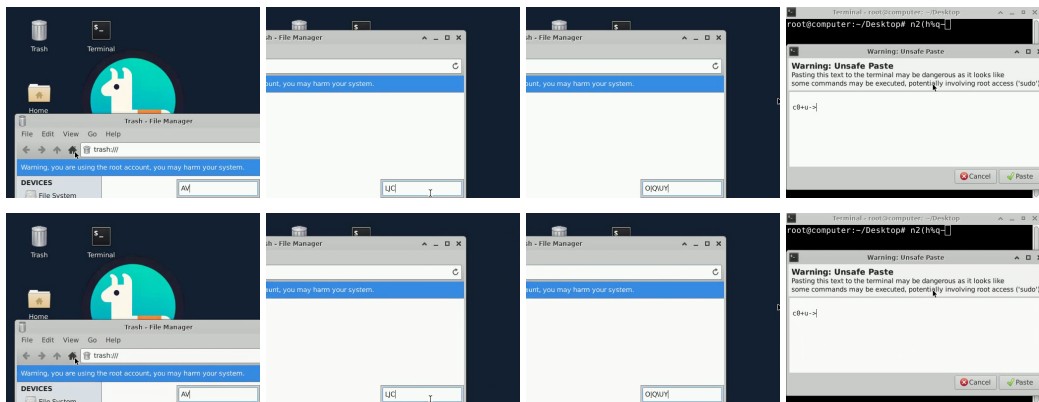

Figure 12: Examples of original images (top row) and their corresponding reconstructions (bottom row) from the trained autoencoder. Despite significant spatial compression ($8\times$ downsampling), the autoencoder preserves details.

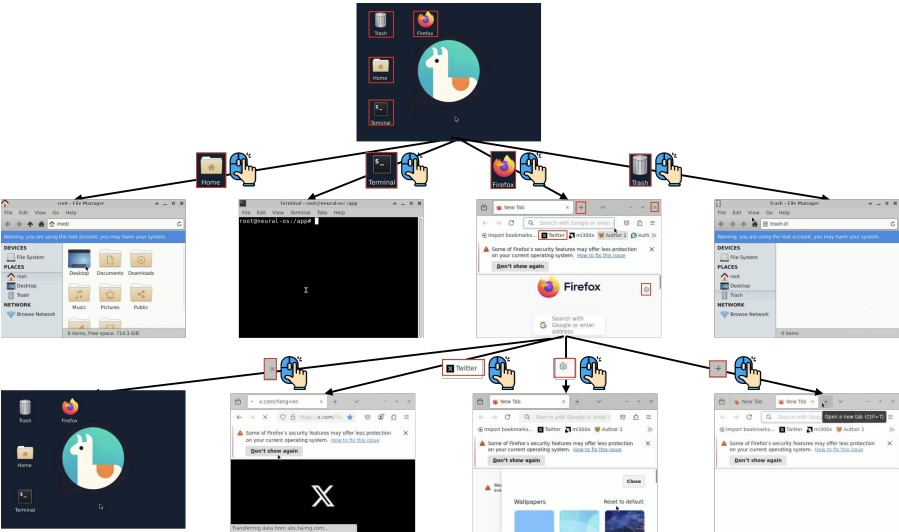

Figure 13: Illustration of Search-Tree-Based Data Collection. We construct a search tree representing OS states, starting from the initial desktop screen (root node). Each node corresponds to a distinct OS state, created by clicking or double-clicking interactable GUI elements identified by a computer-use agent. For clarity, only first-level transitions (opening applications) and one deeper exploration within Firefox are shown. This approach enables collecting diverse interaction data efficiently.

After training, the encoder was able to compress each $512 \times 384$ RGB frame into a latent representation of dimension $16 \times 64 \times 48$ (downsampled by a factor of 8 in spatial dimensions), significantly reducing memory requirements for subsequent NeuralOS model training.

Examples of original and reconstructed images are provided in Figure 12.

## J  COMPUTER-USE AGENT FOR DATA COLLECTION

To build the search tree illustrated in Figure 13, we used structured prompts to guide the computer-use agent. Starting from the initial desktop screen (root node), we sequentially instructed the agent to first map and then verify all interactable GUI elements (Figures 14 and 15). For subsequent GUI states (non-root nodes), we initially prompted the agent to transition to each new state (Figure 16).

```
I need to generate some training data now. Please iteratively map out every application
icon/button on the desktop screen by moving your mouse to it. For each icon, move your
mouse EXACTLY to its center point. DO NOT click it though. Be thorough and identify every
button or icon on the screen from the top to the bottom.

{{suffix}}
```

Figure 14: **Initial GUI Element Mapping (Root Node).** Prompt instructing the agent to identify interactable GUI elements on the initial desktop screen.

```
Now do a final check: Look carefully at every part of the screen for any unmapped buttons.
If you find any, map their exact centers like before. If you are absolutely certain ALL
buttons have been mapped, respond with ONLY the final coordinate list.

{{suffix}}
```

Figure 15: **Final Verification of GUI Elements (Root Node).** Prompt instructing the agent to perform a final check for any missed interactable GUI elements on the initial desktop screen.

```
I need to generate some training data now. First, please click/open the object at {{x}},
{{y}}. Do not progress until it is open or clicked. Double click to open if necessary. Once
you confirmed it is open or clicked, stop here.
```

Figure 16: **Transitioning to a New UI State (Non-Root Node).** Prompt instructing the agent to transition to the current state.

```
Now, iteratively map out all the buttons by moving your cursor to them. Focus on those
related to the object you click/open at {{x}}, {{y}} – especially new buttons. For each
button, move your mouse EXACTLY to its center point. DO NOT click it though.

{{suffix}}
```

Figure 17: **Initial Mapping of Newly Revealed GUI Elements (Non-Root Node).** Prompt instructing the agent to identify GUI elements newly revealed after transitioning to the current UI state.

```
Please continue mapping out buttons on the screen but DO NOT click anymore buttons. Again,
focus on mapping those related to the object you click/open at {{x}}, {{y}} – especially
any new ones. As you exhaust the new buttons, move farther out. For each button, move your
mouse EXACTLY to its center point. DO NOT click it though.

{{suffix}}
```

Figure 18: **Continued Mapping of Remaining GUI Elements (Non-Root Node).** Prompt instructing the agent to further map any remaining interactable GUI elements in the current UI state.

After transitioning, we issued follow-up prompts to map and verify any newly revealed GUI elements (Figures 17 to 19).

Each prompt included a standardized suffix, as shown in Figure 20, to ensure that the agent outputs the coordinates and dimensions of mapped GUI elements in a consistent, structured format. This allowed efficient parsing and automated processing of collected data.

## K   CURSOR POSITION PREDICTION MODEL TRAINING

To quantitatively evaluate cursor position accuracy in the generated frames (Figure 4b), we trained a regression model to predict cursor coordinates directly from screen images. The training procedure is detailed below.

```
Now do a final check: Look carefully around the screen to see if there are any new unmapped
buttons related to the object you click/open at {{x}}, {{y}}. If you find any, map their
exact centers like before. DO NOT click more buttons though. If you are absolutely certain
ALL related buttons have been mapped, respond with ONLY the final coordinate list.

{{suffix}}
```

Figure 19: **Final Verification of GUI Elements (Non-Root Node).** Prompt instructing the agent to perform a final check ensuring all interactable GUI elements have been identified at the current UI state.

```
If you find a keyboard input box, use the type tool specifying its center coordinates and
input box type via text from the following possibilities: terminal, browser, or other, with
coordinates appended in the format x:y.

At the end of your response, provide a structured list of ALL buttons you mapped using this
exact format:

For each button provide two pairs of numbers:
1. Location (L): x~y coordinates where you moved the mouse
2. Size (S): x:y dimensions of the button

Example format:
[
    L100~200 S50:30,  // Button 1: Located at (100,200) with size 50x30
    L300~400 S60:40   // Button 2: Located at (300,400) with size 60x40
]

Important:
- Use EXACTLY this format: Lx~y Sx:y
- Separate each button with commas
- Include ALL buttons you mapped
- Numbers only, no text or descriptions
```

Figure 20: **Structured Output Suffix.** A standardized suffix appended to all prompts, guiding the agent to report mapped GUI elements using a consistent coordinate and dimension format.

**Model Architecture** We used a ResNet-50 convolutional backbone pretrained on ImageNet, with modifications for fine-grained spatial localization tasks. Specifically, we adjusted the stride and dilation parameters in the final convolutional layers to reduce downsampling from $32\times$ to $16\times$, preserving more spatial resolution. The feature extractor output is passed through an additional intermediate convolutional layer followed by a fully-connected regression head, outputting continuous cursor coordinates $(x, y)$.

**Training** We used the Adam optimizer with an initial learning rate of $6 \times 10^{-5}$ and weight decay set to $1 \times 10^{-5}$. We clipped gradients at a maximum norm of 1.0. We optimized an L1 loss between predicted and ground-truth cursor positions. We trained with a batch size of 16 for a total of 2 epochs. Input images were used directly from collected data at the original resolution ($512 \times 384$ pixels), normalized and rearranged to match the input format of ResNet-50. The training data consisted of randomly sampled frames from the full dataset, each labeled with the ground truth cursor positions captured during data collection. Training was performed on a single NVIDIA A6000 GPU. The test error is 0.5 pixels for both x and y. Given that each image is $512 \times 384$ pixels, this reflects extremely high localization precision. In other words, the regression model can predict the cursor location from a screen image with an average deviation of less than a single pixel, making it highly sensitive to small differences and suitable for evaluating fine-grained cursor accuracy in generated frames.

## L USE OF LLMS

We used large language models (LLMs) in two ways. First, we used an LLM-based computer-use agent (Claude-3.5-Sonnet) to generate interaction traces, which were incorporated into our training dataset alongside synthetic data. Second, we used an LLM-based writing assistant to polish

grammar. All ideas, analyses, experiments, and scientific claims are our own, and we take full responsibility for the content of this work.

## M    SCHEDULED SAMPLING IMPLEMENTATION DETAILS

Scheduled sampling requires generating model-based frames during training, which incurs higher computational costs compared to using only ground-truth frames. In a multi-GPU training setting, naively performing scheduled sampling at random intervals could result in synchronization bottlenecks, as some GPUs would have to wait for others to complete computationally expensive sampling steps. To mitigate this issue, we implemented scheduled sampling at regular intervals across all GPUs simultaneously. Specifically, for a scheduled sampling probability of $p = 0.05$, each GPU (and all examples within each GPU's batch) performs scheduled sampling exactly once every 20 steps. This synchronization approach ensures consistent training speed and prevents slowdown due to inter-GPU blocking.

## N    HUMAN EVALUATION

To evaluate how realistic NeuralOS appears to human observers, we conducted a perceptual study modeled after GameNGen (Valevski et al., 2024). Participants were shown pairs of short video clips—one generated by NeuralOS and one recorded from a real Ubuntu XFCE desktop—corresponding to the same underlying sequence of user interactions. Their task was to identify which clip came from the real operating system.

We evaluated six interaction durations: 10s, 20s, 30s, 40s, 50s, and 60s. To control for artifacts that could reveal the model's identity (such as flickering free-space counters, a known issue in diffusion models), the study was conducted under two conditions: (1) full-frame videos and (2) videos cropped to remove the bottom 40 pixels (10% of the height).

Each participant completed 30 randomized trials per condition. A screenshot of the evaluation interface is provided in Figure 21, and quantitative results are reported in Table 1 (main text). In the cropped condition, participants' accuracy fell close to chance, suggesting that NeuralOS simulations are visually convincing for short interaction sequences.

## O    LONG-TERM DEPENDENCY CHALLENGES IN OS SIMULATION

Modeling long-term dependencies is a central challenge in NeuralOS, and it arises in two different ways: (1) practical limitations on context window size during training, and (2) the intrinsic difficulty of maintaining long-horizon state in the model.

**Practical Constraints on Context Window Size**   Increasing the temporal context window substantially increases GPU memory usage and decreases training throughput. To keep training feasible, most stages of NeuralOS training use a 32-frame context window, which allows the model to observe more target transitions per unit of compute. Only in the final stage we extended the context to 64 frames for fine-tuning.

This staged approach mirrors common practice in large language model training, where shorter contexts are used for most of training and longer contexts are introduced only at the end to maintain efficiency. Although longer contexts are possible in principle, they incur prohibitive cost when used throughout training.

**Long-Horizon State Retention Beyond the Context Window**   Even with extended input context, many OS interactions require reasoning over events that happened far earlier (e.g., whether a folder was created hundreds of frames ago). Ideally, longer-term information should be encoded and propagated through the model.

Our folder-presence experiment (Table 2) illustrates this distinction: NeuralOS can recall with a decent accuracy whether a folder was created even after 256 frames, well beyond the 64-frame context used during training. This demonstrates generalization but also shows the broader challenge:

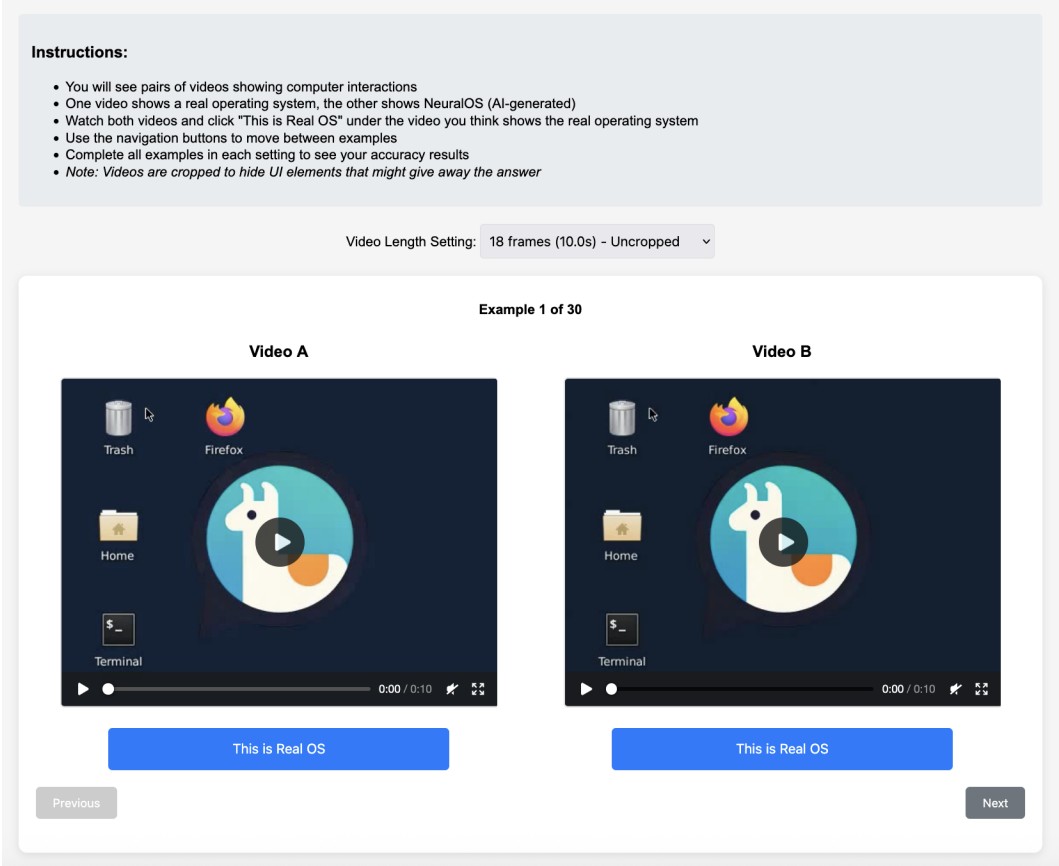

Figure 21: Screenshot of the evaluation interface. Participants were shown side-by-side video clips of the same user interaction sequence, one generated by NeuralOS and one recorded from a real operating system, and asked to identify the real operating system.

for fully interactive OS simulation, models ideally need to preserve relevant state across arbitrarily long horizons.

## P    RELATED WORK

Existing work has proposed neural operating systems in which a generative model mediates between users and the operating environment. Prompt-to-OS (Tolomei et al., 2023) introduces a visionary paradigm where generative models sit between the user and the kernel, potentially replacing traditional application-specific interfaces. We see NeuralOS as a concrete step toward that vision on the user-interaction side: our model learns how screens evolve from low-level input events and could in the future be extended with external tools (e.g., HTTP APIs or storage). Recent systems such as Gemini OS (Shin et al., 2025) and Lambda's NeuralOS (Lambda Cloud, 2025) take a code-generation approach, where user actions are converted into prompts and the model generates HTML or structured UI elements that are rendered by a browser or existing UI engine. In our testing, this approach works well for structured interfaces composed of predefined UI elements (e.g., buttons, panels, grids), but it does not extend to arbitrary pixel-based applications or rich visual content, such as real-time games.

Video generation models such as VideoGPT (Yan et al., 2021) and MoCoGAN (Tulyakov et al., 2017) demonstrated early progress in synthesizing temporally coherent videos. More recent diffusion-based systems, including Imagen Video (Ho et al., 2022a), Make-A-Video (Singer et al., 2022a), Open-Sora (Zheng et al., 2024), and VideoGen (Li et al., 2023), produce high-quality video clips from text prompts or reference frames. However, these approaches differ fundamentally from

NeuralOS: they generally operate by generating video segments using temporally compressed latent representations, rather than performing action-conditioned next-frame prediction. This temporal compression makes them unsuitable for interactive simulation, where each frame must directly depend on fine-grained cursor and keyboard inputs at step-level resolution.

NeuralOS is closely related to recent generative modeling approaches for simulating interactive environments conditioned on user inputs. GameGAN (Kim et al., 2020) used generative adversarial networks (GANs) for interactive game imitation, and Genie (Bruce et al., 2024) generated playable 2D platformer worlds. More recently, diffusion-based models have emerged as powerful real-time simulators: GameNGen (Valevski et al., 2024) simulated the game Doom, MarioVGG (Protocol, 2024) simulated Super Mario Bros, DIAMOND (Alonso et al., 2024) simulated Atari and Counter-Strike, GameGen-X (Che et al., 2024) simulated open-world games, and Matrix (Feng et al., 2024) simulated AAA games. Beyond video games, UniSim (Yang et al., 2023) developed simulators for real-world scenarios.

More broadly, NeuralOS relates to the growing body of work on world models. Early work such as World Models (Ha & Schmidhuber, 2018b) demonstrated predictive simulation for reinforcement learning environments, and subsequent models have scaled this paradigm to large multimodal settings, natural-language control, and tool-augmented agents (LeCun, 2022; Meo et al., 2024; Micheli et al., 2023; 2024; Zhang et al., 2023; Liu et al., 2023). Recent systems such as Pandora (Xiang et al., 2024), Genie-3 (Parker-Holder & Fruchter, 2025), PAN (Xiang et al., 2025), V-JEPA-2 (Assran et al., 2025), Emu (Cui et al., 2025), and Marble (World Labs, 2025) explore increasingly open-ended interactive simulation, often guided by text. World models have also been used as closed-loop training environments for agents (Wu et al., 2023; Ouyang et al., 2022; Ye et al., 2021; Schrittwieser et al., 2020; Alonso et al., 2024; Zhou et al., 2024; Hafner et al., 2023), showing that learned simulations can support planning and skill acquisition.

Compared to these prior works, NeuralOS addresses challenges unique to OS simulation: while most GUI frame transitions involve subtle changes, accurately modeling critical discrete events, such as opening applications or menus, is essential. Additionally, precise cursor position prediction is crucial for interactive usability, NeuralOS introduces targeted model and training innovations specifically addressing these challenges, paving the way toward fully generative OS simulations.

