# OpenReview forum: "NeuralOS: Towards Simulating Operating Systems via Neural Generative Models"
_ICLR.cc/2026/Conference — ICLR 2026 Poster_

### Official Review · Reviewer_YGuk · 2025-10-31

**Soundness:** 4
**Presentation:** 4
**Contribution:** 4
**Rating:** 8
**Confidence:** 4

**Summary:**

This paper suggests that it may be possible to simulate computer programs without having their code through a demonstration of using RNNs + Attention + Diffusion Models with next-pixel-prediction to generate realistic responses to computer input (i.e., keyboard or mouse input).  They show through several experiments that this is highly effective in some toy settings.   They also included evidence that this was not nearly as effective without several novelties added to the process, such as their unique way of representing (x,y) coordinates and their specialized training schedule.

**Strengths:**

This paper is clearly in line with ICLR's goal, to advance theoretical and empirical knowledge of learning.  It has huge potential implications for both theory and practice, suggesting a) that in a very concrete and systematizable way, neural attention mechanisms can replace symbolic logic, and b) suggesting alternatives to current practice in software development.  While I imagine that these results are limited by the simplicity of the environments that were being simulated, that in itself could become an area of theoretical interest in itself - how do we create complexity classes of software applications which can predict whether or not they can be imitated using such a model?

**Weaknesses:**

This paper potentially presents significant ethical challenges - if a model can be trained to in effect "become" a software package, then it will become difficult to preserve the current economic approach to selling software.

Besides that - I understood why they chose to rely on an RNN, but those are famously not as expressive as transformers, and, in general, I would have liked to see more analysis of what kinds of applications \textit{couldn't} be simulated, if any; if literally any software could be simulated, I would have liked to know how they got so much expressivity out of an RNN.  What I would imagine based on their architecture is that the system would be limited in both how complex the software they simulated could be, as well as by how many different software packages a single module could simulate.  I would have liked to see the performance as a function of how many distinct software packages were being imitated by a given module.

**Questions:**

On line 209 - could you please explain the logic for concatenating the noisy image?

Do you have intuition why your unique spatial encoding did better than just including the x,y coordinates?  I would have guessed that the network could have learned to pay particular attention to these details and to tiny differences in those coordinates without requiring a new map formalism (assuming they were floats passed into the module).

Should we think of the scheduled sampling as a kind of teacher forcing, or could you elaborate on theoretical motivations for using it?

On line 265, where it states “After pretraining, the RNN-generated frames alone tend to be blurry due to averaging multiple plausible outcomes, but crucially provide a strong initialization for subsequent joint training” - could you be a bit more specific about the strengths and weaknesses of every stage of training, perhaps in the form of a small table?

Line 270 - I'm surprised occasionally replacing the input image with the model-generated frame works at all if they are actually different and the output crucially depends on what the image is; it seems that if you give it a different input, it should produce a different output as \textit{correct} behavior.  Could you please clarify?

You said that you were using "random exploration" to mine traces, but your search tree looks nothing like random exploration - did you just mean random in the sense of non-deterministic, or am I misunderstanding the role of the search tree in gathering data?

Line 313 - is the use of Bezier curves to model cursor usage novel?  If not, the literature should be cited.  If so, a simple ablation with using a simpler family of curves would be great (although probably more of a camera-ready than necessary-by-rebuttal) item.

**Details Of Ethics Concerns:**

While I don't think the authors acted unethically at all, they are, like so many in the field, introducing software that has the capability to be used for evil (in this case, for software piracy).

---

> ### Author Response · Authors · 2025-12-03
>
> ## Re: Ethical considerations
> Thank you for bringing up this point. We agree that generative models capable of imitating software raise broader questions about intellectual property, distribution models, and the economics of software, much like large language models raised similar concerns. This is the reason why we trained NeuralOS on Ubuntu XFCE environment to avoid copyright infringements. Also, we view NeuralOS as an early research prototype exploring the feasibility of generative interfaces, not as a system intended to replicate or distribute commercial software. Based on your comments, we have added an ethics statement section to our revised draft.
>
>
> ## Re: Expressivity and what types of software could be simulated
>
> Thank you for raising this excellent point. We agree that understanding which kinds of software can or cannot be simulated, and how model capability scales with the diversity and complexity of applications, is an important open direction. Your suggestion points toward a scaling-law-like direction: training models on increasing numbers of applications (1, 2, 4, 8, 16, ...) and measuring when performance begins to saturate, how this depends on software complexity or similarity, and how model capacity must grow accordingly. Unfortunately, such experiments are far beyond our current academic compute budget, but we believe they represent a valuable line of future work enabled by our open release.

---

> > ### Author Response · Authors · 2025-12-03
> >
> > ## Re: Q1 - Logic for concatenating the noisy image
> > Thank you for the question. The diffusion UNet is trained as a denoiser: its input needs to include the noisy version of the target frame at a given diffusion timestep. We concatenate this noisy frame with the RNN state and other conditioning signals so that the model can jointly (1) denoise the latent image and (2) condition on the action and system state. This follows standard diffusion model practice and ensures the UNet receives both the latent dynamics and the corrupted image it must denoise.
> >
> > ## Re: Q2 - Why the spatial encoding outperforms raw (x, y) coordinates
> > We appreciate this insightful question. Passing (x, y) as two scalars places the burden on the renderer to infer where in 2D space to attend, effectively treating cursor location as a 1-of-(512x384) classification problem. In contrast, the spatial cursor map provides a localized Gaussian "spotlight"’ directly in a 2D image. This makes the conditioning spatially aligned with the convolutional structure of the UNet and dramatically improves precision in cursor rendering.
> >
> > Empirically, Figure 4(b) shows that without this spatial map, the cursor often appears in incorrect locations.
> >
> > ## Re: Q3 - Is scheduled sampling "teacher forcing"?
> >
> > Scheduled sampling addresses the train-test distribution mismatch problem used when doing teacher forcing training. During training the model sees ground-truth past frames, but during inference it must condition on its own generated frames, which may contain small errors that compound over time. This problem is also observed in other video generation works such as GameNGen.
> >
> > Scheduled sampling exposes the model to its own predictions during training, allowing the model to learn to fix its own mistakes hence reducing error accumulation. Our Appendix A shows that DIAMOND degrades quickly without this mechanism, while scheduled sampling improves stability both qualitatively (Figure 7) and quantitatively (Figure 8).
> >
> > ## Re: Q4 - Strengths/weaknesses of each training stage
> >
> > Thank you for the suggestion. We added a new table (Table 4) summarizing the purpose, strengths, and remaining limitations of each of the four training stages. This clarifies why each stage exists and how they progressively address failure modes (blurry RNN outputs, exposure bias, short context modeling, etc.).
> >
> > ## Re: Q5 - Why scheduled sampling works despite providing "wrong" inputs
> >
> > This is an important intuition question. Scheduled sampling replaces only the most recent frame, not the entire history. By Stage 3, the model's predictions are already very close to the ground-truth frame, so feeding back a model-generated frame does not significantly shift the target distribution.
> >
> > Replacing more than one frame would cause divergence (e.g., creating a situation where an app opened earlier in the generated history than in the target), but our design of only replacing the last frame mitigates this. Thus, scheduled sampling introduces enough noise to teach robustness while staying within the correct behavior manifold.
> >
> > ## Re: Q6 - Random exploration vs. the search tree
> >
> > We collected data from two different sources:
> >
> > 1. Random exploration: Cursor positions and clicks were sampled randomly across the screen, producing diverse, nondeterministic traces independent of application logic (we did add many heuristics to maximize data diversity and coverage as detailed in our code).
> >
> > 2. Agent data: A computer-use agent explored the OS via a search tree to cover meaningful UI states.
> >
> > By "random exploration" we meant data from the first source.
> >
> > ## Re: Q7 - Bezier curves for cursor modeling
> >
> > Bezier curves were chosen because they provide a compact, smooth parameterization that closely matches human cursor motion. Simpler alternatives (e.g., linear interpolation) produced unrealistic, jittery trajectories. Bezier curves have been used before in trajectory modeling and control; we have added a citation to prior work.

---

### Official Review · Reviewer_9rkZ · 2025-11-01

**Soundness:** 4
**Presentation:** 4
**Contribution:** 2
**Rating:** 6
**Confidence:** 4

**Summary:**

This paper introduces NeuralOS, a generative framework designed to simulate operating system graphical user interfaces by autoregressively predicting screen frames based on user inputs like mouse movements, clicks, and keyboard events. The architecture combines a hierarchical recurrent neural network to manage and track the internal system state with a diffusion-based neural renderer that generates the final screen image conditioned on the RNN's state. The authors ask an LLM-based computer-use agent (Claude-3.5-Sonnet) to generate interaction traces, and alongside random synthetic traces through a Docker Ubuntu instance to generate the underlying training data with actions.

**Strengths:**

1. Cleanly split state tracking (hierarchical RNN with attention over the previous frame) from image synthesis, which makes the problem well-posed for long, interactive sequences.

1. This paper is very instructive in providing a recipe for training video based world-models in general. There are several important tricks here that are more broadly applicable than training a neural OS world model. For instance, the model architecture for long memory, multi-stage training, combination of tricks in Section 4, sampling _transitions_ more often, adding random explorations to mitigate spurious correlations etc.

1. Fast 2-step DDIM giving ~18 fps on a single H100, which is useful for interactive demos, not just offline generation.

1. Reproducibility & openness with code, checkpoints, data scripts, and an interactive demo is linked.

**Weaknesses:**

1. You can't really use the final artifact for much. If you want to train a model to use an operating system, you'd rather just use the Docker container. I'm finding it a bit hard to justify the contribution of this paper beyond that it is a very interesting demo and some of the tricks used to collect the data and make the world-model work.

1. The demonstration paths through the Docker OS to collect training data were generated by a computer-use agent (Claude-3.5-Sonnet). This is obviously very expensive to collect a very large amount of interaction traces.

1. The quality of the model isn't that particularly great. Notably, in the demo, typing into the terminal barely works.

**Questions:**

1. Were the human evaluations done on the same distribution of interaction traces used to collect the data?

1. Given that a standard Docker container provides a perfect, zero-cost, high-fidelity environment for training an OS agent, what is the practical justification for doing 23,000-GPU-hours to train this model? What tangible use case does this simulation serve today the real environment doesn't solve more effectively and cheaply?

1. The paper admits that "fine-grained keyboard interactions are not reliably supported". How can this be presented as a successful OS simulation if it fails at one of the most basic and critical components of an operating system (the terminal)?

1. How exactly does the "fabricated" Doom demo, which was manually constructed from "spliced in" gameplay, demonstrate a generalizable capability to simulate new or useful applications? Isn't this just a one-off, manually-guided demo rather than an emergent property of the model?

---

> ### Author Response · Authors · 2025-12-03
>
> ## Re: Contribution and practical use
>
> Thank you for raising this point. We agree that the current NeuralOS system is not a practical replacement for a real operating system yet. Our goal in this work is different: we aim to demonstrate that fully generative, pixel-level OS simulation is *feasible*, identify the unique modeling challenges involved, and provide a reproducible recipe and open resources for the community.
>
> We believe that, in the long term, human-computer interfaces may increasingly be generated by models rather than manually engineered. A generative OS offers several advantages that conventional OS cannot provide:
>
> 1. Flexibility and learning interfaces from demonstrations
>
> When the interface is generated rather than constructed from predefined widgets, it is no longer constrained to fixed menus, fixed components, or predefined interaction logic. This opens up the possibility of learning UI behavior from demonstrations instead of programming. For example, a designer could illustrate or sketch the intended behavior of an application (e.g., using Photoshop or Nano Banana), and a generative OS could learn that behavior without requiring hand-written UI logic. Our Doom example illustrates this potential: once provided with demonstrations, the model learns a new application interface even though the application is not actually running on the host OS. In principle, a future generative OS could “install” arbitrary applications, even games designed for other operating systems, by being shown demonstrations.
>
> 2. Shared parameters across applications
>
> Traditional operating systems maintain separate storage and rendering pipelines. A unified generative model, by contrast, could share parameters across visually or physically similar environments (e.g., the movie Zootopia and a game built from the same scenes), amortizing both development and storage costs. This parallels the broader shift in AI from many specialized models to a single shared foundation model in language and vision.
>
> 3. Enabling new directions in controllable UI research
>
> Because the interface is generated rather than fixed, a generative OS enables new research directions that are difficult or impossible with traditional systems, such as on-the-fly UI personalization based on user preferences, and text- or speech-conditioned interface transformations (“make my OS look like liquid glass style”)
>
> We are not claiming NeuralOS achieves these capabilities today, but rather that a generative OS offers the flexibility necessary to explore them. We expect continued research and engineering progress, community contributions (supported by our full open-source release), and hardware advances to reduce cost and improve quality. As one example, we have already increased inference resolution from 512x384 to 1024x768 in a newer prototype (now deployed in the demo) without increasing latency too much, largely due to a more efficient autoencoder and inference improvements.
>
>
> ## Re: Cost of using a computer-use agent for data collection
> Thank you for raising this concern. We agree that, in principle, generating large-scale demonstrations with an autonomous computer-use agent could be expensive. In practice, however, this was not a significant cost factor in our work, for several reasons:
>
> 1. Agent-generated traces formed only a small portion of the dataset. The vast majority of our training data comes from random exploration traces which required no LLM usage.
>
> 2. The agent was mainly used to build a search tree for discovering branches of interaction (menus, paths, application launch sequences). The cost of building the tree is amortized across traces we collect after the tree has been built.
>
> ## Re: Demo quality
> Thank you for the observation. NeuralOS is an early research prototype focused primarily on modeling coherent GUI dynamics such as window navigation, cursor behavior, application launches, and menu transitions, etc. Finegrained typing is challenging as it involves rapid, pixel changes and precise keyboard timing for our academic compute budget.
>
> For context, our training budget is significantly smaller than that of related work. For example, Google’s GameNGen trained a simulator for a single application (Doom) using 128 TPUs for 700K steps with 70M examples, whereas NeuralOS attempts to model an entire operating system with many applications at a fraction of that compute. We expect terminal behavior to improve with larger models, bigger datasets, longer training, and community contributions enabled by our full open release.

---

> > ### Author Response · Authors · 2025-12-03
> >
> > ## Re: Q1 - Human evaluation distribution
> > The human evaluation was not conducted on the same distribution as the training data. NeuralOS is trained mostly on synthetic traces generated by random explorations, along with a small set of demonstrations from a computer-use agent. In contrast, the human evaluation uses collected human interaction sequences. These traces reflect real user behavior (e.g., natural cursor motion, hesitation, multi-step exploration patterns), which could differ from the agent-generated traces. We evaluated on this human-collected distribution to test whether NeuralOS generalizes beyond the synthetic behavior used during dataset construction.
> >
> > ## Re: Q2 and Q3 - Justification vs using a Docker container and practical use
> > Thank you for these questions. We have responded to them in "Re: Contribution and Practical Use” above, where we explain our intended research contribution, long-term motivation, and advantages of generative OS that do not arise from a conventional container environment.
> >
> > ## Re: Q4 - Doom demo and generalization to new applications
> > Thank you for this thoughtful question. The Doom example illustrates that a generative OS can learn a new application purely from visual demonstrations, even when that application is not actually running inside the underlying OS. This shows an advantage of pixel-level generative interfaces: new interactive behaviors can be learned from demonstrations rather than manually programmed. In combination with recent advances in image-based UI editing tools (e.g., Nano Banana), we believe this opens up new research directions for designing UIs through demonstration rather than code.
> >
> > We agree that true zero-shot or text-conditioned integration of entirely unseen applications is an ambitious long-term goal. Achieving that would require paired instruction-UI demonstrations and substantially more training compute, which we view as promising future work beyond the scope of the current work.

---

### Official Review · Reviewer_oGA3 · 2025-11-02

**Soundness:** 2
**Presentation:** 3
**Contribution:** 3
**Rating:** 2
**Confidence:** 3

**Summary:**

This paper proposes a framework to simulate operating systems’ (OS) graphical user interfaces (GUIs) through single screen frame prediction in response to user’s inputs, using deep neural networks. The proposed approach spans the entire pipeline - from data collection to the final prediction - and employs two neural architectures: a recurrent neural network (RNN) for state-tracking, and a diffusion-based convolutional neural renderer for image generation. The authors formalized the task of next frame prediction as an autoregressive generative modeling problem.
The models are trained on synthetic data, utilizing both random and AI-agent human-like generated interactions within Ubuntu XFCE environments. The methodology is clearly stated and well presented. The authors address OS-specific simulation challenges - distinct from those encountered in, for example, game environments. Their proposed solutions include notable design choices, such as the use of an RNN for long-term state tracking, and a Gaussian spatial map for precise cursor modeling.
Although no formal metrics or benchmarks are mentioned, the authors report that, based on  human-evaluation results, the model achieves visually coherent frame prediction.
The paper’s main contribution lies in demonstrating that the model can even generalize to previously unseen applications (e.g., clicking an icon for a non-installed app and producing plausible launch behavior). One suggestion regarding this last point, would be to expand on the significance of this finding and its broader implications for the future of generative user interfaces. The work is motivated by the goal of creating learned, interactive simulators of operating systems through synthetic demonstrations, but the research motivation behind this should be stated more clearly.

**Strengths:**

* The paper is clearly written and well organized; the technical components are formally defined and adequately justified, which makes the work easy to follow.
* The proposed architectures are coherently structured and integrated.
* The paper represents an original contribution, as generative simulation of Operating Systems remains novel and unexplored, though it draws some parallels with prior work on generative simulation in other fields (i.e.: gaming).
* Code is given to reviewers, allowing the reproducibility of the proposed approach.

**Weaknesses:**

* The motivation remains vague throughout the paper. The key questions left to be answered are: “How can this contribution advance the field?”, “Why do we need to simulate OSs?”. A more explicit problem statement - together with expected applications (i.e.: human-computer interaction research, AI agents) would strengthen the paper’s significance.
* No closely related previous work is discussed - hence, no competitors are referenced. The authors only cite examples from game or real-time simulations, omitting directly relevant efforts in neural or generative OS simulations. Providing a stronger contextualization of the work, would help in collocating the study - e.g.: by referencing recent work such as “Simulating a Neural Operating System by Google (at https://developers.googleblog.com/en/simulating-a-neural-operating-system-with-gemini-2-5-flash-lite/) or other generative interfaces models (as in  https://arxiv.org/pdf/2310.04875). This would clarify how the proposed approach advances beyond prompt-driven systems. * This too, would aid the authors in the articulation of why a fully autonomous OS would be desirable.
* Experimental validation is limited. Although ablation studies are included to assess each components’ importance, no quantitative comparison against state-of-the-art baselines is provided. Qualitative evaluations and human-evaluation results are present and informative, but they would benefit from additional quantitative evidence (i.e.: comparison with SOTA, statistical analyses, benchmark comparisons, graphs).
* Development is limited to synthetic Ubuntu XFCE environments, hence, the generalization ability to other OS interfaces remains to be verified.
* Regarding the provided demo, the simulation exhibits some technical issues: frame rendering lags; disappearing cursor behavior when the RNN toggle is activated; the “Auto Input” option not working - there is no frame prediction (i.e.: the screen does not change) and after about five seconds of waiting for an autonomous screen update, one gets a “Connection Timeout Warning” and the connection restarts. These issues make it difficult to evaluate the system’s claimed interactivity.
* An additional concern regards the resources required. The high GPU cost required for the proposed OS simulation - which cannot approach performance comparable to real-time OSs - raises questions about utility and scalability. In simpler words: “Why is it necessary to use all those hours of GPU usage in order to simulate an OS that will never reach performance comparable to a real OS (which is also, way cheaper)?”

**Questions:**

See weaknesses

---

> ### Author Response · Authors · 2025-11-27
>
> ## Re: Motivation, problem statement, and expected applications.
> Thank you for this suggestion. Based on your feedback, we have revised the introduction to add a problem statement and expected applications. In the updated introduction, we have added the following paragraph:
>
> > In this work, we study whether an operating system-like interface can be modeled using a neural generative model, rather than a manually programmed system. In doing so, we investigate the modeling and optimization challenges, such as precise cursor control and state tracking. Such a neural interface provides a basis for human-computer interaction research on user-controllable UI generation (e.g., prompt-to-UI personalization), and provides a safe environment in which computer-use agents can be trained and evaluated without issuing real system commands.
>
> ## Re: Related work and contextualization
> Thank you for raising this point. Based on your feedback, we have expanded the Related Work section to discuss Prompt-to-OS (P2OS) and recent systems like Gemini OS and Lambda's NeuralOS. In the revised text, we clarify that P2OS proposes a broader vision in which generative models mediate between the user and the kernel, and we position NeuralOS as a concrete step toward this vision on the user-interaction side. We also contrast NeuralOS with code-generation approaches like Gemini OS and Lambda's NeuralOS, which generate HTML or structured UI elements that are rendered by a browser or existing UI runtime. By comparison, NeuralOS operates at the pixel level, enabling it to generate arbitrary visual content (e.g., our Doom example).
>
> Concretely, we have added the following paragraph to the paper:
> > Existing work has proposed neural operating systems in which a generative model mediates between users and the operating environment. Prompt-to-OS (P2OS) introduces a visionary paradigm where generative models sit between the user and the kernel, potentially replacing traditional application-specific interfaces. We see NeuralOS as a concrete step toward that vision on the user-interaction side: our model learns how screens evolve from low-level input events and could in the future be extended with external tools (e.g., HTTP APIs or storage). Recent systems such as Gemini OS and Lambda's NeuralOS take a code-generation approach, where user actions are converted into prompts and the model generates HTML or structured UI elements that are rendered by a browser or existing UI engine. In our testing, this approach works well for structured interfaces composed of predefined UI elements (e.g., buttons, panels, grids), but it does not extend to arbitrary pixel-based applications or rich visual content, such as real-time games like Doom.
>
> We hope this revision better contextualizes our work.
>
> ## Re: Experimental validation
> Thank you for this suggestion. While we are not aware of existing models designed specifically for pixel-level operating-system simulation, we adapted the DIAMOND work ("Diffusion for World Modeling"), originally developed for video games such as Atari and CS:GO, as the closest comparable baseline with open-source implementation. In response to your feedback, we added a new appendix section (Appendix A) comparing DIAMOND and NeuralOS under matched compute and data.
>
> To enable a fair comparison, we extended DIAMOND's context window size from 4 to 64 frames, number of keys from 11 to 179, and increased cursor coordinate resolution from 23x17 bins to full 512x384 resolution to suit finegrained GUI interaction (such as clicking on small buttons). We also trained a scheduled-sampling variant of DIAMOND. As shown in the new qualitative comparison figure (Figure 7), both DIAMOND variants struggle with abrupt frame transitions such as application launches and degenerate, while NeuralOS is more stable. We also report pixel-level RMSE over generation length (Figure 8), where NeuralOS maintains lower error across longer time spans. For convenience, we include a summary of the RMSE (normalized to 0-1) versus the number of frames generated here:
>
> | # Frames | DIAMOND | DIAMOND + Scheduled Sampling | NeuralOS |
> | -------: | ------: | ---------------------------: | -------: |
> |        1 |  0.05 |                       0.05 |   0.05 |
> |        2 |  0.28 |                       0.09 |   0.05 |
> |        4 |  0.44 |                       0.11 |   0.05 |
> |        8 |  0.51 |                       0.29 |   0.05 |
> |       16 |  0.55 |                       0.43 |   0.06 |
> |       32 |  0.57 |                       0.48 |   0.08 |
> |       64 |  0.54 |                       0.49 |   0.14 |
> |      128 |  0.51 |                       0.45 |   0.21 |
>
> These results show the challenges specific to OS simulation (e.g., long-term state tracking and precise cursor modeling), which differ from the assumptions commonly made in game-based simulation, where short context windows suffice as most game states (ammunition, health) are visually observable from recent frames.

---

> ### Author Response · Authors · 2025-11-27
>
> ## Re: Evaluation beyond Ubuntu XFCE
> Thank you for pointing this out. We agree that evaluating whether NeuralOS generalizes across different operating systems is an important direction for future work. Our use of Ubuntu XFCE in the current work reflects a practical constraint rather than a limitation of the methodology. Since our goal is to release all data, models, and code openly, we selected a system that is fully redistributable without licensing restrictions. Additionally, collecting and training comparable datasets for multiple operating systems would significantly increase storage and compute requirements, which was beyond the scope of our available academic budget.
>
> To make this clearer, we have updated the limitations section (Appendix B) with the following text:
> > The current study uses Ubuntu XFCE as the operating environment. This choice reflects practical rather than technical constraints: Ubuntu provides a legally redistributable platform aligned with our goal of releasing all resources openly, and training comparable models across multiple operating systems would substantially increase data collection, storage, and compute requirements. Extending NeuralOS to more operating systems such as Windows or macOS remains an important direction for future work.
>
> ## Re: Demo behavior and interactive settings
> Thank you for taking the time to test the demo. We appreciate your careful interaction and trying features beyond the default configuration.
>
> Some of the behaviors you observed relate to internal debugging options that were exposed in the interface and are not intended for normal use. For example, the “Use RNN” toggle bypasses the diffusion model and directly decodes the RNN state. This mode is only valid during the early RNN-pretraining stage (where RNN outputs were trained to match latent frames), and after joint training with the diffusion model it no longer produces valid frames (as the objective has changed to be diffusion loss), hence the degraded visual output. Similarly, "Auto Input" is not an autonomous control or agent mode. NeuralOS only generates the next frame when it receives a new input event, unlike a conventional operating system where the screen keeps refreshing. Without repeated input, the model would simply hold the last frame indefinitely even if the user is waiting and expecting the UI to continue evolving. Auto Input therefore repeatedly sends the last cursor position at a fixed interval to drive the model forward through delayed visual transitions (such as opening Firefox).
>
> The connection timeout warning you encountered is a backend resource policy rather than a model limitation. Since each session occupies a GPU worker, idle timeouts and session limits are used to avoid one user occupying the GPU for too long to allow other users to use it.
>
> To avoid confusion, we have now:
> - Hidden the advanced debugging options by default, while still making them accessible via a developer console command `showDebugControls()` for users who want to explore internals.
> - Added to Appendix F detailing the full demo implementation.
> We appreciate the reviewer’s careful testing and feedback!

---

> ### Author Response · Authors · 2025-11-27
>
> ## Re: GPU cost and practical utility
>
> Thank you for raising this question! We agree that the current system is not yet a practical replacement for a conventional operating system. We view the present work as an early research prototype rather than an optimized or deployment-ready solution. Our goal in this paper is to explore whether an OS interface can be modeled generatively and to surface the challenges involved.
>
> We believe a generative OS may offer two long-term advantages that traditional OS architectures cannot provide:
> - Flexibility: Because the interface is generated at the pixel level rather than built from predefined UI components, it is not constrained by existing GUI toolkits. Our Doom demonstration illustrates this potential: if demonstrations exist, the system can learn entirely new interactive behaviors without manually designing UI elements. This suggests a future where UI behavior could be demonstrated rather than implemented. For example, a designer sketches or edits an interaction in tools like Photoshop or Nano Banana, and a generative OS learns and implements that UI.
> - Shared parameters across applications: Today, applications such as video games and movies each maintain their own storage and rendering pipelines. A single model capable of modeling multiple applications may amortize storage cost, especially when different applications share structure (e.g., similar scenes in different video games and movies and physical rules). While speculative, this mirrors how single large multi-purpose models are replacing many single-purpose pipelines (such as machine translation and document summarization) in language processing.
>
> We also expect efficiency to improve in the future. Continued engineering efforts, community contributions (facilitated by our fully open release), and hardware advances should all reduce cost over time. As an example of iterative improvement, we have already increased the operating resolution from 512x384 to 1024x768 in a newer prototype (now already deployed in the public demo to demonstrate this point) without increasing latency by much, largely due to a better autoencoder and inference optimizations (such as by only sending image regions that changed).
>
> That said, we fully agree that compute remains a current limitation. To make this explicit, we have updated the limitations section with the following text:
> > For inference, the current system requires an H100 GPU and is not yet suitable for cost-efficient or practical deployment. This reflects the exploratory nature of the work. We expect future architectures, training strategies, and hardware to make such systems substantially more efficient.
>
> We hope this clarifies the motivation of our work. Rather than arguing that a generative OS is immediately practical, our objective is to establish a basis for open research in this direction and to provide resources enabling the community to explore and improve.

---

### Official Review · Reviewer_cFb2 · 2025-11-02

**Soundness:** 3
**Presentation:** 4
**Contribution:** 3
**Rating:** 6
**Confidence:** 3

**Summary:**

This paper proposes NeuralOS, an end-to-end generative model–based operating system capable of simulating and rendering graphical user interfaces (GUIs) through real-time cursor and keyboard interactions. The study adopts a multi-stage training framework to enhance the quality of GUI simulation. Experiments and demo results suggest that NeuralOS can effectively simulate GUIs via GPU rendering without relying on a physically hosted interface. Overall, this work presents an intriguing step toward developing a world model for GUI simulation.

**Strengths:**

This work provides a clear and detailed description of the model architecture, input feature encoding, and multi-stage training strategy to ensure rendering quality. The use of LLMs to autonomously collect interaction data, thereby removing human involvement, is particularly interesting. Overall, the paper demonstrates a promising approach to constructing a world model for operating systems, capable of generating real-time screens based on user interactions.

**Weaknesses:**

#1 The paper should better position its work within the context of existing works. For instance, although video generation is briefly mentioned (line 146), no specific papers are cited. Similarly, despite of the discussion on world models in Section L, the main text does not contextualize NeuralOS in world models. The authors clarify how their approach to modeling long-term trajectories and user actions differs from prior methods used in video generation and world modeling.

#2 The discussion of challenges in lines 214–215 is unclear. In Stage 4, the paper notes that capturing long-term dependencies is difficult due to hardware limitations, yet the input context is later extended to address this. If this extension mitigates the issue, the authors should clarify in what sense this remains a challenge.

#3 The paper mentions that transformers suffer from high inference complexity, yet the proposed model incorporates attention heads which is an essential component of transformers.

#4 Cursor coordinates are encoded twice, at line 174 and line 199, respectively. The paper should clarify the rationale behind this encoding design or conduct an ablation study to explain how it contributes to the simulation of cursor movement.

#5 In evaluating the performance of NeuralOS, the performance metrics and their formulas should be clarified and referenced. For example, the formula for accuracy is not explicitly provided in the main text. It is unclear whether the accuracy reported in Table 1 (which measures the identification of the real system) is conceptually the same as the cursor position accuracy and overall image simulation accuracy.

**Questions:**

Please answer the questions in the weakness section.

---

> ### Author Response · Authors · 2025-12-03
>
> ## Re: Positioning relative to video generation and world models
> Thank you for this suggestion. In the revised draft, we have expanded the Related Work section:
>
> **Video generation:** We added a paragraph discussing prior video generation models and clarifying why these methods are not directly comparable to NeuralOS. Specifically, we added the following text to the Related Work section:
>
> > Video generation models such as VideoGPT and MoCoGAN demonstrated early progress in synthesizing temporally coherent videos. More recent diffusion-based systems, including Imagen Video, Make-A-Video, Open-Sora, and VideoGen, produce high-quality video clips from text prompts or reference frames. However, these approaches differ fundamentally from NeuralOS: they generally operate by generating video segments using temporally compressed latent representations, rather than performing action-conditioned next-frame prediction. This temporal compression makes them unsuitable for interactive simulation, where each frame must directly depend on fine-grained cursor and keyboard inputs at step-level resolution.
>
> **World models:** We also added a paragraph discussing world models:
>
> > More broadly, NeuralOS relates to the growing body of work on world models. Early work such as World Models demonstrated predictive simulation for reinforcement-learning environments, and subsequent models have scaled this paradigm to large multimodal settings, natural-language control, and tool-augmented agents. Recent systems such as Pandora, Genie-3, PAN, V-JEPA-2, Emu, and Marble explore increasingly open-ended interactive simulation. World models have also served as training environments for agents, demonstrating that learned simulations can support planning and skill acquisition.
>
> **Positioning NeuralOS within world models:**
> We have discussed the difference between NeuralOS and existing world models in Related Work, in terms of the modeling challenges unique to OS simulation:
>
> > Compared to these prior works, NeuralOS addresses challenges unique to desktop operating-system simulation: many GUI frame transitions involve very subtle visual changes, yet discrete events such as opening applications or menus must be modeled with high fidelity; and precise cursor-position prediction is essential for interactive usability. NeuralOS introduces architectural and training strategies specifically targeting these challenges, extending world-model techniques to a domain where long-term state tracking and pixel-level precision are critical.
>
> We hope these additions better contextualize NeuralOS.
>
> ## Re: Clarification on long-term dependency challenges (lines 214–215)
> If we understood correctly, your concern is that the paper states that long-term dependencies are difficult to model due to computational constraints, yet later in Stage 4 we extend the context window from 32 to 64 frames. This may appear contradictory unless we clarify in what sense long-term dependencies remain a challenge even after the context extension.
>
> To address this, we have added a new appendix section (Appendix O: Long-Term Dependency Challenges in OS Simulation) explaining the issue more precisely. In short, the difficulty arises in two ways:
>
> **Training-time practical constraints:**
>  Extending the temporal context window increases GPU memory usage and reduces training throughput. For this reason, most of NeuralOS’s training uses a 32-frame context, and we only expand to 64 frames during the final fine-tuning stage. This mirrors common practice in large language model training, where long contexts are introduced only at the end due to cost. Thus, it is not that large contexts are impossible, but that using them throughout training is expensive and impractical.
>
> **State retention beyond the context window:**
>  Many OS tasks require remembering events far beyond 64 frames. For example, if a folder was created hundreds of frames ago, we don't want the model to forget that. Even with extended context, the model must retain relevant information in its hidden states. Our folder-presence experiment demonstrates partial success (e.g., a decent accuracy of recalling the presence of a folder after 256 frames), but also shows that perfect long-horizon memory remains a core challenge for generative OS simulation.

---

> > ### Author Response · Authors · 2025-12-03
> >
> > ## Re: Use of attention vs. transformer inference complexity
> > Thank you for pointing this out. When we referred to the high inference complexity of transformers, we were specifically referring to full-sequence self-attention, where at inference time each new token (or frame) attends to all previously generated tokens. Without truncation, this causes inference cost per step to grow linearly with sequence length, making it unsuitable for long-horizon, real-time OS simulation.
> >
> > In contrast, the attention mechanism used in NeuralOS is structurally very different. Our model uses single-step cross-attention from the current RNN state to the most recent frame only. The model does not attend over the entire interaction history; long-term information is instead maintained in the recurrent state.
> >
> > Therefore, the complexity issue we mentioned is not due to the presence of attention itself, but due to the full-sequence usage pattern characteristic of transformers.
> >
> > ## Re: Cursor coordinates appear to be encoded twice
> > Thank you for this careful observation. Cursor coordinates indeed enter the model through two separate pathways, but the two encodings serve different purposes in different components of the architecture.
> >
> > **(1) Encoding for the RNN (action modeling):**
> > In the first instance, the raw cursor (x,y) coordinates are embedded using positional encodings and added to the keyboard-action embedding before being fed into the RNN. This allows the RNN to reason about where user actions occur (e.g., distinguishing a click on Firefox from a click on Terminal). Also, more importantly, during Stage 1 (RNN pretraining), the UNet is not used at all, so this embedding is the only way for the RNN to perceive cursor location. Without it, the RNN would not be able to distinguish a click on one button from a click on another button.
> >
> > **(2) Encoding for the UNet (rendering):**
> > The second encoding constructs a Gaussian cursor map in the latent image space, which is concatenated with the RNN output before UNet rendering. This allows rendering the cursor at the correct pixel location. As shown in Figure 4(b), removing this spatial cursor map leads to significant degradation in cursor position rendering in the generated frames.
> >
> > Therefore, the cursor coordinates were passed once to RNN to allow it to understand what actions are performed, and another time to the UNet to allow it to render cursor at the correct position.
> >
> > ## Re: Clarification of performance metrics and use of the term accuracy
> > Thank you for raising this point. We agree that the term accuracy was previously used in several different contexts which could lead to confusion. In the revised manuscript, we have clarified each metric and ensured that the terminology is unambiguous:
> >
> > - Human study (Table 1): we have modified the caption as “Human success rate (%) of identifying the real OS.”
> > - Folder-presence experiment (Table 2): we have modified the caption to refer to this as a “folder presence confusion matrix”.
> >
> > We have also clarified in the main text what metrics mean when we first introduce them in the revised draft.

---

### Author Response · Authors · 2025-12-03
**General Response to the AC**

Thank you for taking the time to evaluate our submission. Below we summarize the main concerns raised by reviewers and how our revised draft and rebuttal address them.

## 1. Strengthening positioning within the literature

Multiple reviewers requested clearer connections to prior work in video generation, world models, and prompt-based UI/code generation.

**Revisions made**

* Added a new paragraph on video generation, explaining why temporally compressed video models are not suitable for action-conditioned next-frame OS simulation.
* Added a new paragraph situating NeuralOS within the broader world-modeling literature.
* Expanded the discussion of code-generation OS systems (e.g., Gemini OS), clarifying how NeuralOS's pixel-level generative approach differs from methods that output HTML/DOM structures.


## 2. Adding a baseline from the literature

Some reviewers noted the absence of a comparable baseline. During the rebuttal period, we implemented and trained a DIAMOND (Diffusion for World Modeling) baseline adapted to our setting.

**Revisions made**

* Added a new Appendix A comparing NeuralOS with DIAMOND, originally used for Atari and CS:GO simulation.
* Adapted DIAMOND to OS simulation with matched compute budget, action space, input resolution, and context window.
* Added qualitative comparisons (Figure 7) and an RMSE plot (Figure 8) showing the unique challenges in OS simulation compared to video game simulators.

## 3. Improving the technical presentation

Reviewers requested clarification on evaluation metrics, training stages, context-length constraints, cursor encodings, demo behavior, and ethical considerations.

**Revisions made**

* Added an explicit problem statement in the Introduction.
* Added Table 4 summarizing the purpose and limitations of each training stage.
* Clarified the rationale behind the two cursor encodings (RNN-level vs. renderer-level).
* Added Appendix F detailing the demo implementation and modified the public demo so debugging controls are hidden by default.
* Clarified all evaluation metrics (human study, folder-presence test, cursor accuracy).
* Added Appendix O explaining why long-term dependencies remain challenging even after context extension.
* Added an Ethics Statement addressing concerns around generative replication of software.

## 4. Addressing concerns about motivation and practicality

Several reviewers asked about the utility of a generative OS compared to simply running a traditional OS.

**Revisions made**

We clarified that NeuralOS is an early open-source research prototype aimed at exploring the feasibility of generative interfaces, not a practical OS replacement yet. Our goals include:

* Demonstrating that OS-level UI behavior can be learned generatively rather than hard-coded, and surfacing the technical challenges (precise cursor modeling, abrupt frame transitions, long-term state tracking, etc).
* Showing that UI behavior can be learned from demonstrations (as in the Doom example), opening the possibility of demonstration-driven interfaces.
* Opening up long-term research directions such as online UI personalization and parameter sharing across applications (e.g., games and movies), which are difficult to achieve under traditional software architectures.
* Releasing the full data collection pipeline, model checkpoints, training code, and demo implementation so the community can reproduce, build on, and extend this research direction.

We appreciate the reviewers' and area chair's time and thoughtful evaluations of our work.

---

### Meta-Review · Area_Chair_Ksdx · 2026-01-07

**Summary:**

reviewers gave 2,6,6,8, with the main negative reviewer highlighting the following weaknesses:

Motivation: vague, and not clear why we need to simulate OSs. needs a more explicit problem statement together with expected applications (i.e.: human-computer interaction research, AI agents). also lack of related work discussion

Experimental validation: Although ablation studies are included to assess each components’ importance, no quantitative comparison against state-of-the-art baselines is provided. Qualitative human-evaluation results are good but lack of quantitative evidence (i.e.: comparison with SOTA, statistical analyses, benchmark comparisons, graphs).

Development is limited to synthetic Ubuntu XFCE environments, generalization to other OS interfaces remains to be verified.

Provided demo faces some engineering and technical issues.

Concerns about GPU resources required.

**Reviewer Concerns:**

Motivation: vague, and not clear why we need to simulate OSs. needs a more explicit problem statement together with expected applications (i.e.: human-computer interaction research, AI agents). also lack of related work discussion.

--> several reviewers raised this issue, I think the authors have addressed them.

Experimental validation: Although ablation studies are included to assess each components’ importance, no quantitative comparison against state-of-the-art baselines is provided. Qualitative human-evaluation results are good but lack of quantitative evidence (i.e.: comparison with SOTA, statistical analyses, benchmark comparisons, graphs).

--> added some quantitative results by adapting the DIAMOND work ("Diffusion for World Modeling") to evaluate video-game/OS world generation. seems to be a good addition to the paper.

Development is limited to synthetic Ubuntu XFCE environments, generalization to other OS interfaces remains to be verified.

--> mostly attributed to future work

Provided demo faces some engineering and technical issues.

--> addressed, also i feel this is a minor issue

Concerns about GPU resources required.

--> mostly attributed to future work

**Reviewer Scores:**

oGA3 who gave 2 likely to increase to 4. other reviewers positive 6 6 8

---

### Decision · Program_Chairs · 2026-01-26

Accept (Poster)